# Assessing risk factors and impact of cyberbullying victimization among university students in Myanmar: A cross-sectional study

**Aye Thazin Khine[1,2], Yu Mon Saw[2,3]\*, Zaw Ye Htut[1], Cho Thet Khaing[4], Htin Zaw Soe[5], Kyu Kyu Swe[5], Thinzar Thike[6], Hein Htet[7], Thu Nandar Saw[8], Su Myat Cho[2], Tetsuyoshi Kariya[2,3], Eiko Yamamoto[2], Nobuyuki Hamajima[2]**

**1** Department of Public Health, Ministry of Health and Sports, Nay Pyi Taw, the Republic of the Union of Myanmar, **2** Department of Healthcare Administration, Nagoya University Graduate School of Medicine, Nagoya, Japan, **3** Nagoya University Asian Satellite Campuses Institute, Nagoya, Japan, **4** Department of Epidemiology, University of Public Health, Yangon, the Republic of the Union of Myanmar, **5** University of Community Health, Magway, the Republic of the Union of Myanmar, **6** Department of Food and Drug Administration, Ministry of Health and Sports, Nay Pyi Taw, the Republic of the Union of Myanmar, **7** Department of Preventive and Social Medicine, University of Medicine, Mandalay, the Republic of the Union of Myanmar, **8** Department of Community and Global Health, the University of Tokyo, Tokyo, Japan

\* sawyumon@med.nagoya-u.ac.jp

**Data Availability Statement:** Data are available upon request as a requirement of the Institutional Review Board, University of Public Health, Yangon, Myanmar and Ethics Review Committee, University of Community Health, Magway, Myanmar for

## Abstract

### Background

Cyberbullying is a global public health concern with tremendous negative impacts, not only on the physical and mental health of students but also on their well-being and academic performance. However, there are very few studies on cyberbullying among university students, especially in Myanmar. This study aims to determine the percentage of university students who suffered cyberbullying victimization in the last 12 months, and the association between students' socio-demographic characteristics, adverse events following cyberbullying and cyberbullying victimization.

### Methods

A cross-sectional study was conducted among university students aged 18 years and older at one medical university in Magway, Myanmar. A total of 412 students (277 males and 135 females) participated in the study. Data were collected from August to September, 2018 using a self-administered questionnaire. Multiple logistic regression analyses (models I and II) were performed to estimate the unadjusted (UOR) and adjusted odds ratios (AOR), and 95% confidence intervals (CI).

### Results

In total, 40.8% of males and 51.1% of females in the study had suffered cyberbullying victimization in the past 12 months. In model I, students who had been studying at the university for 3 years or less (AOR = 1.81; 95% CI 1.14–2.85), and who had witnessed psychological, physical or sexual violence, or cyberbullying in their neighborhoods, (AOR = 2.95; 95% CI

researchers who meet the criteria for access to confidential data. Researchers who would like to access to the data must contact Planning Unit, Department of Public Health, Office no. (47), Ministry of Health and Sports, Ottara Thiri Township, Ministry Zone, Nay Pyi Taw 15011, Myanmar. Tel: 95-673-431448, Fax: 95-673-431083. Email: planning.dph@mohs.gov.mm.

**Funding:** The authors received no specific funding for this work.

**Competing interests:** The authors have declared that no competing interests exist.

1.48–5.91) were more likely to have suffered cyberbullying victimization in the past 12 months. In model II, being a victim of cyberbullying was associated with difficulties in concentrating and understanding lectures (AOR = 3.96; 95% CI 1.72–9.11), and substance abuse (AOR = 2.37; 95% CI 1.02–5.49). Non-resident students were at a higher risk of being cyberbullying victims than their resident peers (AOR = 1.86; 95% CI 1.04–3.34).

## Conclusion

Two out of five students had suffered cyberbullying victimization in the past 12 months, and only half of the victims discussed their experience(s) with someone else. Students who suffered cyberbullying victimization faced academic difficulties and started or increased smoking, betel chewing or alcohol drinking. Counter measures to prevent and mitigate the adverse events related to cyberbullying victimization are urgently needed among university students in Myanmar. Periodic screening for cyberbullying, counseling services, cyber-safety educational programs, and awareness raising campaigns should be implemented.

## Introduction

Smart phones with internet usage have increased worldwide making communication, information sharing, and learning or applying new technologies easier within and across countries. The global advances in the use of this technology has had many positive impacts on society in the 21st century. On the other hand, several negative consequences brought by this technology are inevitable, and cyberbullying is one of them [1–4].

Cyberbullying can be regarded as "a new violence type of the era" [1] especially among school children, adolescents, and youths [2–4]. It is also considered a hidden epidemic. It can have numerous adverse effects on people from all walks of life, and owing to the involvement of technology, it can occur any time [3–5]. Cyberbullying is defined as "an aggressive and intentional act that is carried out using electronic forms of contact by a group or an individual repeatedly and over time against a victim who cannot easily defend him or herself" [6]. Cyberbullying characteristics are similar to traditional bullying, which is an act or behavior which is intentional, aggressive and repetitive in nature, causing harm to a victim such that the victim finds it difficult to defend themselves owing to power imbalance [5]. However, in cyberbullying, the perpetrators can stay anonymous allowing greater potential to do or say more harmful things to the victims than they would in personal relations [7]. Moreover, the perpetrators can reach out to the victim 24 hours a day via internet accessibility and can send annoying mails, messages or spread rumors online which can draw a larger audience than traditional bullying [7].

There are many different ways to perpetrate cyberbullying through internet or smart phone technology, which at its most extreme includes online suicide challenge games [8]. These include hacking or stalking a person's Facebook or social media account(s) or smart phone, and impersonating them, using a person's picture online without his or her consent, telling lies or spreading false rumors about a person behind his or her back, sending humiliating, annoying or mean texts or posts or sex chat, making upsetting phone calls or malicious prank calls, sending unpleasant photos, sexually explicit images or videos to a person without their consent, and taking a photo/photos of a person or videotaping a person without their consent and using the photo(s) or video(s) to humiliate or threaten them [6,9,10].

Numerous studies have been conducted to assess the status of cyberbullying among middle and high school students; however, only few have focused on university students [4–6,11–18]. Several studies conducted among undergraduates on the prevalence of cyberbullying have found that it ranges from 10% to 60%, across different countries [1,11–13,17,18]. The differing prevalence of cyberbullying may be due to the different operational definitions for cyberbullying, different methodologies such as classification depending on frequencies (at least once or several times, etc.) or reported time interval (within one week, one month, one year, or lifetime, etc.), and different socio-demographic characteristics of the study participants in various studies [7]. Astonishingly, in most studies, the rate of reporting by victims was observed to be very low compared with the prevalence of cyberbullying [6,15,19–23].

School children who suffered cyberbullying victimization in their adolescence were more likely to encounter the same situation as undergraduates and in their adulthood [2–4]. Moreover, cyberbullying and traditional bullying were found to be interrelated [24–26]. Students who had witnessed or faced different forms of bullying or violence in their surroundings were more likely to suffer cyberbullying victimization [1,6,15,16,19,23]. Cyberbullying was also associated with the amount of time spent on the internet or using social media [6,27]. The more time spent online, the higher the risk of being victimized by cyberbullying [6,27]. University students who live separately from their families may experience loneliness or isolation [28–31]. These students may end up spending more time on the internet and using social media to combat this isolation or loneliness, which increases their risk of becoming cyberbullying victims [6,27,30,31].

Overwhelmingly, the evidence indicates that cyberbullying is associated with poor physical and mental health and academic performance [23,26,28,29,32–37]. Students who have suffered cyberbullying victimization are more likely to experience emotional stress, anxiety, or depression [23,32–34]. When the situation worsens, they may harbor thoughts of suicide or even attempt suicide [33,34]. Cyberbullying causes victims to feel unsafe and distressed, which in turn affects their interest in their studies and worsens their academic performance [26,28,29,35–37]. In addition, being cyberbullying victims was positively associated with increased substance abuse (e.g., smoking, alcohol abuse or drug addiction) [23,24,38].

To surf the worldwide information technology tide, widespread internet and smart phone usage is inevitable in many countries making technology consumers more prone to cyberbullying. Although cyberbullying is common among middle school students [7,39], more than 30% of undergraduate students reported that they first experienced cyberbullying in college [40], and an equal victimization rate was found between male and female students [41]. Cyberbullying is very common in South East Asian (SEA) countries [42]. Studies related to cyberbullying in SEA countries observed 59.4% of cyberbullied victims among Facebook users in Singapore [43], 39.7% of young-adult (17 to 30 years) victims in Malaysia [44], 59% of cyberbullying victims in Thailand [45] and 80% of junior high school students experiencing cyberbullying victimization in Indonesia [46].

Today, smart phone internet usage is also ubiquitous among university students in Myanmar. University students who are in transitional phase of life into adulthood are willing to try and learn new things. Widespread smart phone internet usage makes it easier for students to stay current, and even actively involved, in the things that interest them. With the use of this technology comes several pros and cons. Effective and efficient interventions are urgently needed to control undesirable aspects of the information technology tide such as cyberbullying. However, Myanmar being a developing country lags behind in the protection of technology consumers against cyberbullying. The first cyber law in Myanmar is still in the early stages of development. Currently, protection against cyberbullying is indirectly provided under the Telecommunications Law and the Electronic Transactions Law.

These laws only provide an ambiguous protection, leaving the student population at risk. There is a lack of statistical data assessing current cyberbullying situation among university students in Myanmar [42]. Therefore, this study aimed to determine the percentage of university students who suffered cyberbullying victimization in the past 12 months, and the associations between students' socio-demographic characteristics, adverse events following cyberbullying and victimization.

## Methods

### Study area and participants

The participants in this study were male and female university students aged 18 years and older in their second to final year, and from the condensed health assistant course during the 2018–2019 academic year, at a medical university in Magway, Myanmar. The condensed health assistant (CHA) course is different from the regular 4-year health assistant course. This 9-month condensed course is intended for the public health staff of the Department of Public Health (DoPH), Ministry of Health and Sports (MoHS), Myanmar. The participants of CHA course started their career as Public Health Supervisor Grade 2 (PHS 2) in DoPH, MoHS. When they have at least 3 years government service, they can sit for the promotion exam for Public Health Supervisor Grade 1 (PHS 1). After 3 years of service as PHS 1, they get another chance to sit for the entrance exam to join the CHA course. Therefore, the age of the students in the CHA course is much older than the students in regular 4-year health assistant course. The participants who attended the lectures at the day of data collection and who gave their written informed consent to participate in the study were recruited. The total number of students attending the 2018 academic year at the university was 802. Among them, 453 students gave their consent to participate in this study. After data cleaning, 41 participants were excluded from the dataset due to incomplete data. Finally, 412 students (277 males and 135 females) were included in the data analysis.

### Data collection

This cross-sectional study was conducted from August to September 2018 over a period of two months. The data were collected with a pre-tested, self-administered questionnaire. The aim of the study and questionnaire contents were explained to the participants prior to obtaining their consent. Data collection was conducted only after obtaining written informed consent from the participants.

The questionnaire was developed by referring to the domestic violence questionnaire from the 2015–2016 Myanmar Demographic and Health Survey [47], and the gender-based violence factsheet published by the Department of Public Health, Ministry of Health and Sports, Myanmar [9]. The questionnaire was pretested among 30 university students at one university other than medical universities in Magway. The pre-test results were used to finalize the questionnaire. The questionnaire consisted of three sections: 1) socio-demographic characteristics of the participants, 2) experiences with regard to cyberbullying victimization, and 3) association with adverse events following cyberbullying victimization.

Socio-demographic characteristics and adverse events following cyberbullying victimization were considered as independent variables, and having suffered cyberbullying victimization in the past 12 months as the dependent variable.

### Operational definition of cyberbullying

In this study, cyberbullying was defined as the following aggressive or intentional acts involving internet or smart phone technology: hacking or stalking a person's Facebook or social

media account(s) or smart phone and impersonating them; using a person's picture online without his or her consent; telling lies or spreading false rumors about a person behind his or her back; sending humiliating, annoying or mean texts, posts or sex chat; making upsetting phone calls or malicious prank calls; sending unpleasant photos, sexually explicit images or videos to a person without their consent; and taking a photo/photos of a person or videotaping a person without their consent and using the photo(s) or video(s) to humiliate or threaten them [6,9].

## Statistical analysis

The data were analyzed using SPSS version 25 software. The socio-demographic characteristics, participants' experiences with regard to cyberbullying victimization and the adverse association of cyberbullying were described using frequency and percentage for each sex. Multiple logistic regression analyses (models I and II) were performed to estimate the unadjusted (UOR) and adjusted odds ratios (AOR) and 95% confidence intervals (CI). The significant value of p was considered as less than 0.05.

## Ethical considerations

Ethical approval was obtained from the Institutional Review Board, University of Public Health, Ministry of Health and Sports, Myanmar (Ethical No. UPH-IRB [2018/Research/27] issued on 31st July 2018) and Ethics Review Committee, University of Community Health, Magway, Myanmar (Ethical No. UCH-ERC/09/2018 issued on 6th July 2018). Data collection was conducted only after obtaining written informed consent from the participants. Confidentiality of all collected data was ensured at each and every stage of data handling. The data were anonymized using a random identity number at the start of data collection.

## Results

Table 1 presents the socio-demographic characteristics of the participants according to sex. Among the 412 university students, majority (77.4%) of both male and female students were aged between 18 and 22 years at the time of the survey. About one-third of the participants (35% males and 26.7% females) had been studying at the university for four years or more. It was observed that 80.3% of the students came to Magway from different states and regions of Myanmar. Almost all the students (98.8%) used Facebook. Approximately half of the students (54.9% of males and 48.1% of females) spent more than two hours per day on social media. Most of the students (83.8% of males and 92.6% of females) reported that they had witnessed psychological or physical or sexual violence or cyberbullying in their neighborhoods.

Data on the participants' experiences with regard to cyberbullying victimization in the past 12 months are described in Table 2. Of all the participants, 44.2% (40.8% in males and 51.1% in females) reported they suffered cyberbullying victimization in the past 12 months (p = 0.048). The majority of both male (77.9%) and female (65.2%) victims had experienced one or more of the types of cyberbullying victimization mentioned above by age 20 or younger.

Table 3 describes the adverse association of cyberbullying victimization among the victims. Of the male victims, 26.5% started or increased smoking, betel chewing (smokeless tobacco), or alcohol drinking (p<0.001), and 29.2% found it more difficult to concentrate and understand lectures than usual after having suffered cyberbullying victimization in the past 12 months (p = 0.023). Among female victims, 7.2% considered attempting suicide after experiencing cyberbullying victimization.

**Table 1. Socio-demographic characteristics of the participants (N = 412).**

| Characteristics | Total (N = 412) | | Males (N = 277) | | Females (N = 135) | |
|---|---|---|---|---|---|---|
| | N | % | N | % | N | % |
| **Age group [completed years] (mean = 2.67, SD = 1.22)** | | | | | | |
| 18–22 | 319 | 77.4 | 236 | 85.2 | 83 | 61.5 |
| 23–27 | 63 | 15.3 | 29 | 10.5 | 34 | 25.2 |
| 28–35 | 30 | 7.3 | 12 | 4.3 | 18 | 13.3 |
| **Marital status** | | | | | | |
| Single | 386 | 93.7 | 258 | 93.1 | 128 | 94.8 |
| Married | 26 | 6.3 | 19 | 6.9 | 7 | 5.2 |
| **Total year(s) of study (mean = 2.67, SD = 1.22)** | | | | | | |
| ≤1 | 87 | 21.1 | 35 | 12.6 | 52 | 38.5 |
| 2 | 113 | 27.4 | 90 | 32.5 | 23 | 17.0 |
| 3 | 79 | 19.2 | 55 | 19.9 | 24 | 17.8 |
| ≥4 | 133 | 32.3 | 97 | 35.0 | 36 | 26.7 |
| **Residence** | | | | | | |
| Magway | 81 | 19.7 | 61 | 22.0 | 20 | 14.8 |
| Other | 331 | 80.3 | 216 | 78.0 | 115 | 85.2 |
| **Facebook usage** | | | | | | |
| No | 5 | 1.2 | 1 | 0.4 | 4 | 3.0 |
| Yes | 407 | 98.8 | 276 | 99.6 | 131 | 97.0 |
| **Other social media usage[§]** | | | | | | |
| No | 236 | 57.3 | 156 | 56.3 | 80 | 59.3 |
| Yes | 176 | 42.7 | 121 | 43.7 | 55 | 40.7 |
| **Average hour(s) per day spent on social media (mean = 2.88, SD = 1.83)** | | | | | | |
| ≤1 | 78 | 18.9 | 42 | 15.1 | 36 | 26.7 |
| 1 to 2 | 117 | 28.4 | 83 | 30.0 | 34 | 25.2 |
| >2 | 217 | 52.7 | 152 | 54.9 | 65 | 48.1 |
| **Witnessed psychological, physical or sexual violence, or cyberbullying** | | | | | | |
| No | 55 | 13.3 | 45 | 16.2 | 10 | 7.4 |
| Yes | 357 | 86.7 | 232 | 83.8 | 125 | 92.6 |

[§]Other social media usage: usage of Instagram or YouTube or Viber or WeChat or Bee Talk or Twitter

In the unadjusted analysis, having been cyberbullied was strongly associated with difficulty in concentrating and understanding lectures (UOR = 6.81; 95% CI 3.31–13.98) and starting or increasing substance abuse (UOR = 4.57; 95% CI 2.25–9.31). In addition, cyberbullying victims were approximately three times more likely to tell anyone about their experience(s) (UOR = 3.58; 95% CI 2.34–5.47), and to have ever witnessed psychological, physical or sexual violence, or cyberbullying in their neighborhoods (UOR = 3.26; 95% CI 1.66–6.38). Participants who were female (UOR = 1.52; 95% CI 1.00–2.30), older than 21 years of age (UOR = 1.60; 95% CI 1.02–2.51), or had studied at the university for three years or less (UOR = 1.98; 95% CI 1.29–3.05) were more likely to suffer cyberbullying victimization (Table 4).

Tables 5 and 6 show the results of the multiple logistic regression analysis. In model I, socio-demographic characteristics of the cyberbullying victims _ age, sex, marital status, total year(s) of study, residence, Facebook usage, other social media (Instagram, YouTube, Viber, WeChat, Bee Talk, and Twitter) usage, average hour(s) per day spent on social media, and having ever witnessed psychological or physical or sexual violence or cyberbullying in the

**Table 2. Participants' experiences regarding cyberbullying victimization in the past 12 months (N = 412).**

| Characteristics | Total (N = 412) | | Males (N = 277) | | Females (N = 135) | | p-value# |
|---|---|---|---|---|---|---|---|
| | N | % | N | % | N | % | |
| **Suffered cyberbullying victimization** | | | | | | | 0.048 |
| No | 230 | 55.8 | 164 | 59.2 | 66 | 48.9 | |
| Yes | 182 | 44.2 | 113 | 40.8 | 69 | 51.1 | |
| **Impersonation**★ | | | | | | | 0.843 |
| No | 362 | 87.9 | 244 | 88.1 | 118 | 87.4 | |
| Yes | 50 | 12.1 | 33 | 11.9 | 17 | 12.6 | |
| **Being used your pictures online without your consent** | | | | | | | 0.016 |
| No | 343 | 83.3 | 222 | 80.1 | 121 | 89.6 | |
| Yes | 69 | 16.7 | 55 | 19.9 | 14 | 10.4 | |
| **Being told lies or having false rumors spread about you** | | | | | | | 0.299 |
| No | 376 | 91.3 | 250 | 92.4 | 126 | 93.3 | |
| Yes | 36 | 8.7 | 27 | 7.6 | 9 | 6.7 | |
| **Received humiliating, annoying, mean texts or posts or sex chats** | | | | | | | 0.016 |
| No | 354 | 85.9 | 246 | 88.8 | 108 | 80.0 | |
| Yes | 58 | 14.1 | 31 | 11.2 | 27 | 20.0 | |
| **Received upsetting phone calls or malicious prank calls** | | | | | | | <0.001 |
| No | 294 | 71.4 | 214 | 77.3 | 80 | 59.3 | |
| Yes | 118 | 28.6 | 63 | 22.7 | 55 | 40.7 | |
| **Received unpleasant photos, sexually explicit images or videos** | | | | | | | 0.766 |
| No | 345 | 83.7 | 233 | 84.1 | 112 | 83.0 | |
| Yes | 67 | 16.3 | 44 | 15.9 | 23 | 17.0 | |
| **Outing**¶ | | | | | | | 0.330 |
| No | 394 | 95.6 | 263 | 94.9 | 131 | 97.0 | |
| Yes | 18 | 4.4 | 14 | 5.1 | 4 | 3.0 | |
| **First experienced age of cyberbullying victimization among victims (N = 182 [113 males + 69 females]; mean = 19.8, SD = 3.88)** | | | | | | | 0.062 |
| ≤20 | 133 | 73.1 | 88 | 77.9 | 45 | 65.2 | |
| >20 | 49 | 26.9 | 25 | 22.1 | 24 | 34.8 | |

★Impersonation: Someone hacked or stalked and used the participant's Facebook, other social media account(s) or smart phone and pretended to be him/her.

¶Outing: Someone took a photo/photos of the participant or videotaped him/her without his/her consent and used the photo(s) or video(s) to humiliate or threaten him/her.

#A chi-square test for the difference between males and females

neighborhoods _ were included (Table 5). In model II, the adverse events following cyberbullying victimization among the students _ starting or increasing use of substances, experiencing difficulty in concentrating and understanding lectures, considering attempting suicide, and telling anyone about the cyberbullying victimization experience(s) _ were added to the variables in model I (Table 6).

According to model I, students who had been studying at the university for three years or less (AOR = 1.81; 95% CI 1.14–2.85) and those who had witnessed psychological or physical or sexual violence or cyberbullying in their neighborhoods (AOR = 2.95; 95% CI 1.48–5.91) were more likely to suffer cyberbullying victimization in the past 12 months (Table 5).

In model II, cyberbullying victims were nearly four times more likely to face difficulties in concentrating and understanding lectures, in comparison to non-victims (AOR = 3.96; 95% CI 1.72–9.11). Students who told anyone about their experience(s) (AOR = 2.87; 95% CI 1.78–

**Table 3. Adverse association of cyberbullying victimization among the victims (N = 182).**

| Characteristics | Total (N = 182) | | Males (N = 113) | | Females (N = 69) | | p-value# |
|---|---|---|---|---|---|---|---|
| | N | % | N | % | N | % | |
| **Started or increased use of substance¶** | | | | | | | <0.001 |
| No | 148 | 81.3 | 83 | 73.5 | 65 | 94.2 | |
| Yes | 34 | 18.7 | 30 | 26.5 | 4 | 5.8 | |
| **Being more difficult to concentrate and understand lectures** | | | | | | | 0.023 |
| No | 139 | 76.4 | 80 | 70.8 | 59 | 85.5 | |
| Yes | 43 | 23.6 | 33 | 29.2 | 10 | 14.5 | |
| **Considered attempting suicide in the past 12 months** | | | | | | | 0.595 |
| No | 171 | 94.0 | 107 | 94.7 | 64 | 92.8 | |
| Yes | 11 | 6.0 | 6 | 5.3 | 5 | 7.2 | |
| **Told anyone about cyberbullying victimization experience(s)** | | | | | | | 0.039 |
| No | 89 | 48.9 | 62 | 54.9 | 27 | 39.1 | |
| Yes | 93 | 51.1 | 51 | 45.1 | 42 | 60.9 | |

¶substance: smoking or betel chewing (smokeless tobacco) or alcohol drinking

#A chi-square test for the difference between males and females

4.64) and who reported starting or increasing substance abuse (AOR = 2.37; 95% CI 1.02–5.49) were approximately two times more likely to be cyberbullying targets. Students who came to Magway from different states and regions to attend the university (AOR = 1.86; 95% CI 1.04–3.34) and those who were studying at the university for three years or less were more likely to have suffered cyberbullying victimization in the past 12 months (Table 6)."

## Discussion

This study is the first to examine cyberbullying victimization status, and the association between students' socio- demographic characteristics, adverse events following cyberbullying and being victims among university students in Myanmar. This study revealed that having suffered cyberbullying victimization in the past 12 months was positively associated with difficulty in concentrating and understanding lectures, and starting or increasing substance abuse. Only half of the students who were cyberbullied in the past 12 months told anyone about their experience(s). Students who had witnessed psychological or physical or sexual violence or cyberbullying in their neighborhoods were more likely to suffer cyberbullying victimization. In addition, coming from a different state and region to attend the university and having attended the university for three years or less were positively associated with suffering cyberbullying victimization in the past 12 months. In the unadjusted analysis, being female and being older than 21 years of age were associated with increased risk of being cyberbullying victims. Moreover, cyberbullying victimization is associated with having suicidal ideas in the past 12 months.

Cyberbullying has been described as "new bottle but old wine" [15,16]. Many studies have demonstrated that among students, traditional bullying as well as cyberbullying is strongly related to poor academic performance or outcomes [24–26,28,35–37]. Students who suffered peer bullying received lower grades, and faced academic difficulties and/or worsened academic performance compared with their non-bullied peers [24–26,28,29,35–37]. In this context, this study is consistent with the evidence from previous studies [24–26,28,29,35–37]. Cyberbullying victims were nearly four times more likely to face difficulties in concentrating and

**Table 4. Unadjusted analysis of maximum likelihood and odds ratio estimates for cyberbullying victimization among students in the past 12 months (N = 412).**

| Parameter | df | Maximum likelihood estimates | | | | OR estimates | | |
| --- | --- | --- | --- | --- | --- | --- | --- | --- |
| | | Estimate(B) | SE | Wald χ2 | P value | Parame-ter | UOR | 95% CI |
| **Age [completed years] (mean = 2.67, SD = 1.22)** | | | | | | | | |
| >21 | 1 | 0.47 | 0.23 | 4.19 | **0.041** | vs. ≤21 | 1.60 | 1.02–2.51 |
| **Sex** | | | | | | | | |
| Female | 1 | 0.42 | 0.21 | 3.90 | **0.048** | vs. Male | 1.52 | 1.00–2.30 |
| **Marital status** | | | | | | | | |
| Married | 1 | 0.42 | 0.41 | 1.04 | **0.308** | vs. Single | 1.51 | 0.69–3.36 |
| **Total year(s) of study (mean = 2.67, SD = 1.22)** | | | | | | | | |
| ≤3 | 1 | 0.68 | 0.22 | 9.66 | **0.002** | vs. >3 | 1.99 | 1.29–3.05 |
| **Residence** | | | | | | | | |
| Other | 1 | 0.37 | 0.26 | 2.07 | 0.150 | vs. Magway | 1.44 | 0.88–2.38 |
| **Facebook usage** | | | | | | | | |
| Yes | 1 | 0.17 | 0.92 | 0.04 | 0.850 | vs. No | 1.19 | 0.20–7.20 |
| **Other social media usage§** | | | | | | | | |
| Yes | 1 | 0.37 | 0.20 | 3.43 | 0.064 | vs. No | 1.45 | 0.98–2.15 |
| **Average hour(s) per day spent on social media (mean = 2.88, SD = 1.83)** | | | | | | | | |
| 1 to 2 | 1 | -0.40 | 0.30 | 1.83 | 0.176 | vs. ≤1 | 0.67 | 0.38–1.20 |
| >2 | 1 | -0.23 | 0.27 | 0.77 | 0.382 | vs. ≤1 | 0.79 | 0.47–1.33 |
| **Witnessed psychological or physical or sexual violence or cyberbullying** | | | | | | | | |
| Yes | 1 | 1.18 | 0.34 | 11.84 | **0.001** | vs. No | 3.26 | 1.66–6.38 |
| **Started or increased use of substance¶** | | | | | | | | |
| Yes | 1 | 1.52 | 0.36 | 17.56 | **<0.001** | vs. No | 4.58 | 2.25–9.31 |
| **Being more difficult to concentrate and understand lectures** | | | | | | | | |
| Yes | 1 | 1.92 | 0.37 | 27.24 | **<0.001** | vs. No | 6.81 | 3.31–13.98 |
| **Considered attempting suicide in the past 12 months** | | | | | | | | |
| Yes | 1 | 2.69 | 1.05 | 6.57 | **0.010** | vs. No | 14.73 | 1.88–115.20 |
| **Told anyone about cyberbullying victimization experience(s)** | | | | | | | | |
| Yes | 1 | 1.28 | 0.22 | 34.68 | **<0.001** | vs. No | 3.58 | 2.34–5.47 |

§Other social media usage: usage of Instagram or YouTube or Viber or WeChat or Bee Talk or Twitter

¶substance: smoking or betel chewing (smokeless tobacco) or alcohol drinking

understanding lectures compared to non-victims, as bullying creates an unsafe environment for the students to live in [28]. Cyberbullying is sometimes considered to be more extreme than traditional bullying, as it has several unique characteristics that can affect victims anywhere, and at any time, creating small or large negative impacts [17,25,26]. Periodic screening of cyberbullying and counselling services for students who suffered cyberbullying are also needed as these interventions are found to be effective to prevent and mitigate adverse outcomes of cyberbullying [17,18,22,32], and Myanmar still lacks such kind of interventions in schools or universities.

In this study, only half of the cyberbullying victims told anyone about their experience(s). The low percentage of victims who were willing to tell others about their suffering in this study was consistent with the percentages found in the majority of previous studies [6,15,19–23]. There might be several important reasons behind the less reporting observed in this study. The main reason may be the victims' feeling of helplessness in the face of being attacked, or negligence by the adults or authority figures concerning cyberbullying, as there is currently no direct law or regulation against cyberbullying in Myanmar [48,49]. Other reasons for less

**Table 5. Adjusted analysis of maximum likelihood and odds ratio estimates for cyberbullying victimization among students in the past 12 months (N = 412) [Model I✿].**

| Parameter | df | Estimate(B) | SE | Wald χ2 | P value | Parame-ter | AOR | 95% CI |
|---|---|---|---|---|---|---|---|---|
| | | **Maximum likelihood estimates** | | | | **OR estimates** | | |
| **Age [completed years] (mean = 2.67, SD = 1.22)** | | | | | | | | |
| >21 | 1 | 0.18 | 0.28 | 0.44 | 0.510 | vs. ≤21 | 1.20 | 0.70–2.06 |
| **Sex** | | | | | | | | |
| Female | 1 | 0.25 | 0.23 | 1.21 | 0.270 | vs. Male | 1.29 | 0.82–2.02 |
| **Marital status** | | | | | | | | |
| Married | 1 | 0.15 | 0.45 | 0.11 | 0.740 | vs. Single | 1.16 | 0.48–2.83 |
| **Total year(s) of study (mean = 2.67, SD = 1.22)** | | | | | | | | |
| ≤3 | 1 | 0.59 | 0.23 | 6.42 | **0.011** | vs. >3 | 1.81 | 1.14–2.85 |
| **Residence** | | | | | | | | |
| Other | 1 | 0.33 | 0.27 | 1.52 | 0.218 | vs. Magway | 1.39 | 0.82–2.34 |
| **Facebook usage** | | | | | | | | |
| Yes | 1 | 0.62 | 0.95 | 0.43 | 0.514 | vs. No | 1.86 | 0.29–11.91 |
| **Other social media usage§** | | | | | | | | |
| Yes | 1 | 0.19 | 0.22 | 0.73 | 0.392 | vs. No | 1.21 | 0.79–1.86 |
| **Average hour(s) per day spent on social media (mean = 2.88, SD = 1.83)** | | | | | | | | |
| 1 to 2 | 1 | -0.27 | 0.31 | 0.74 | 0.391 | vs. ≤1 | 0.77 | 0.41–1.41 |
| >2 | 1 | -0.30 | 0.29 | 0.01 | 0.918 | vs. ≤1 | 0.97 | 0.55–1.70 |
| **Witnessed psychological, physical or sexual violence, or cyberbullying** | | | | | | | | |
| Yes | 1 | 1.08 | 0.35 | 9.35 | **0.002** | vs. No | 2.95 | 1.48–5.91 |

§Other social media usage: usage of Instagram or YouTube or Viber or WeChat or Bee Talk or Twitter

✿ Model I adjusted for age, sex, marital status, total year(s) of study, residence, Facebook usage, other social media usage, average hour(s) per day spent on social media, and witnessing psychological, physical or sexual violence, or cyberbullying in their neighborhoods.

reporting may be the fear of internet access restriction or prohibition by family or teachers, and facing more cyberbullying attacks if they reported [32]. Therefore, it is important to create and implement awareness raising campaigns and educational programs regarding cyber safety in Myanmar. Legal protection measures against cyberbullying should be formulated and applied within the university setting.

In this study, having been cyberbullied was found to be positively associated with starting or increasing smoking, betel chewing (smokeless tobacco), or alcohol drinking. Similarly, many studies found higher possibility of alcohol, tobacco, or substance abuse among cyberbullying victims as they were more likely to engage in unhealthy or improper behaviors, and more likely to face psychosocial problems [23,24,38]. To address these issues, strong peer support systems including friends, teachers, parents, and family members are needed to fully support the victims both physically and psychologically [4,17]. In addition, victim-blaming should be avoided.

The majority of participants in this study had witnessed at least one type of psychological, physical, sexual violence or cyberbullying in their neighborhoods, and they were more likely to have suffered cyberbullying victimization in the past 12 months. This finding is in line with those of other studies that found that approximately two-thirds or half of the students reported knowing someone who had experienced cyberbullying [6,16], and the majority of victims were found to have experienced more than one form of bullying [38] due to the correlation between traditional bullying and cyberbullying [6,15,19,23].

Nearly 20% of students lived in Magway and most of the respondents were non-residences who came from different states or regions to attend the university. Moreover, it was found that

**Table 6. Adjusted analysis of maximum likelihood and odds ratio estimates for cyberbullying victimization among students in the past 12 months (N = 412) [Model II\*].**

| Parameter | df | Estimate(B) | SE | Wald χ2 | P value | Parameter | AOR | 95% CI |
|---|---|---|---|---|---|---|---|---|
| | | Maximum likelihood estimates | | | | OR estimates | | |
| **Age [completed years] (mean = 2.67, SD = 1.22)** | | | | | | | | |
| >21 | 1 | 0.11 | 0.30 | 0.14 | 0.705 | vs. ≤21 | 1.12 | 0.62–2.03 |
| **Sex** | | | | | | | | |
| Female | 1 | 0.28 | 0.26 | 1.20 | 0.272 | vs. Male | 1.33 | 0.80–2.19 |
| **Marital status** | | | | | | | | |
| Married | 1 | 0.59 | 0.47 | 1.56 | 0.211 | vs. Single | 1.81 | 0.72–4.56 |
| **Total year(s) of study (mean = 2.67, SD = 1.22)** | | | | | | | | |
| ≤3 | 1 | 0.52 | 0.25 | 4.29 | **0.038** | vs. >3 | 1.69 | 1.03–2.77 |
| **Residence** | | | | | | | | |
| Other | 1 | 0.62 | 0.30 | 4.30 | **0.038** | vs. Magway | 1.86 | 1.04–3.34 |
| **Facebook usage** | | | | | | | | |
| Yes | 1 | 1.08 | 1.01 | 1.13 | 0.289 | vs. No | 2.93 | 0.40–21.32 |
| **Other social media usage§** | | | | | | | | |
| Yes | 1 | 0.28 | 0.24 | 1.34 | 0.248 | vs. No | 1.32 | 0.83–2.10 |
| **Average hour(s) per day spent on social media (mean = 2.88, SD = 1.83)** | | | | | | | | |
| 1 to 2 | 1 | -0.11 | 0.34 | 0.11 | 0.743 | vs. ≤1 | 0.89 | 0.46–1.74 |
| >2 | 1 | 0.15 | 0.31 | 0.23 | 0.634 | vs. ≤1 | 1.16 | 0.63–2.14 |
| **Witnessed psychological or physical or sexual violence or cyberbullying** | | | | | | | | |
| Yes | 1 | 0.66 | 0.37 | 3.13 | 0.077 | vs. No | 1.93 | 0.93–4.00 |
| **Started or increased use of substance¶** | | | | | | | | |
| Yes | 1 | 0.86 | 0.43 | 4.04 | **0.044** | vs. No | 2.37 | 1.02–5.49 |
| **Being difficult to concentrate and understand lectures** | | | | | | | | |
| Yes | 1 | 1.38 | 0.43 | 10.45 | **0.001** | vs. No | 3.96 | 1.72–9.11 |
| **Considered attempting suicide in the past 12 months** | | | | | | | | |
| Yes | 1 | 1.32 | 1.10 | 1.46 | 0.227 | vs. No | 3.76 | 0.44–32.22 |
| **Told anyone about cyberbullying victimization experience (s)** | | | | | | | | |
| Yes | 1 | 1.05 | 0.25 | 18.52 | **<0.001** | vs. No | 2.87 | 1.78–4.64 |

§Other social media usage: usage of Instagram or YouTube or Viber or WeChat or Bee Talk or Twitter

¶substance: smoking or betel chewing (smokeless tobacco) or alcohol drinking

*Model II adjusted for age, sex, marital status, total year(s) of study, residence, Facebook usage, other social media (Instagram or YouTube or Viber or WeChat or Bee Talk or Twitter) usage, average hour(s) per day spent on social media, ever witnessing psychological or physical or sexual violence or cyberbullying in the neighborhoods, started or increased use of substance, being difficult to concentrate and understand lectures, considering attempting suicide in the past 12 months, and telling anyone about cyberbullying victimization experience(s).

non-resident students were more likely to suffer cyberbullying victimization compared with their local peers. Non-resident students separated from their families might feel stressed or socially isolated [30,31] and therefore, ended up spending more time on the internet, increasing their risk to suffer cyberbullying victimization [6,27].

Respondents who had studied at the university for three years or less were more likely to have suffered cyberbullying victimization. Studies identifying the association between university years and cyberbullying are still limited. However, previous studies have suggested that cyberbullying experiences can continue into adulthood but occur mostly in adolescents [2–4]. Most of the participants in this study who had attended the university for less than three years were still in their adolescence. They were more likely to be victimized compared to senior

students due to their curiosity to engage in new things which may attract unintentional, adverse events such as cyberbullying.

In the unadjusted analysis, being female and being older than 21 years of age were found to be associated with being a cyberbullying victim. This finding was consistent with those of several studies in which females were found to be more victimized by electronic means than males [6,15,19]. However, a study conducted among Turkish youth observed that both males and females suffered cyberbullying equally [50]. Although cyberbullying was found to occur primarily during adolescence, the same situation can also be experienced as an undergraduate or as an adult [2–4] and the negative consequences suffered by victims can be carried over to their work place [51]. In addition, suicidal ideation was positively associated with cyberbullying victimization in the past 12 months in this study which supported a similar association found in the studies conducted among cyberbullied victims in the United States, France, Hong Kong and several other countries [33,34,52,53]. These studies also observed that cyberbullying victimization is much more likely to be positively associated with suicidal ideation compared to traditional bullying or cyberbullying perpetration alone [33,34,52,53].

Cyberbullying was found to be associated with the amount of time spent on the internet or social media [6,27,54,55]. The longer the time spent online, the higher the risk of suffering cyberbullying victimization [6,27,54,55]. However, no association between the amount of time spent on social media per day among students and suffering cyberbullying victimization in the past 12 months was found in this study. The reason may be due to the difference in the use of common social media among various countries. Majority of participants in this study used Facebook as the most common social media in their daily lives, and cyberbullying attack can occur with one post or comment in Facebook or one message through messenger where the victims do not need to spend much time online.

The cross-sectional nature of this study limited the ability to identify causal relationships between cyberbullying victimization and various risk factors. Subjective and recall bias might have affected the accuracy of the data which have been self-reported. The extent of the adverse association of cyberbullying could not be measured accurately as no measurement scale or a comparison group was used in this study. The age eligibility of study participants and the study being conducted was limited to only one university in Magway, Myanmar limiting the potential generalizability of the results as there are different demographic and socio-economic conditions across the country.

Future studies should examine the gender differences in various types of cyberbullying such as those of perpetrators or bully/victims among university students. Studies exploring the cause-effect relationship between cyberbullying and potential risk factors are recommended. The adverse consequences of cyberbullying should be assessed more accurately in future studies. Studies focusing on preventive as well as mitigation or coping strategies for adverse consequences of cyberbullying especially targeting the adolescents and youth population in Myanmar are also recommended. Gender differences should be considered in formulating these strategies.

## Conclusion

To conclude, two out of five students suffered cyberbullying victimization in the past 12 months. Non-resident students and students who had been studying in the university for three years or less were at a higher risk of being cyberbullying victims. Students who had witnessed psychological, physical or sexual violence, or cyberbullying in their neighborhoods were more likely to experience cyberbullying victimization. Cyberbullying victims were found to be positively associated with difficulty in concentrating and understanding lectures, and starting or

increasing substance abuse. Periodic screening for cyberbullying, counseling services, cyber-safety educational programs, and awareness raising campaigns are urgently needed for university students in Myanmar.

## Supporting information

**S1 Questionnaire. (Myanmar questionnaire).**
(PDF)

**S2 Questionnaire. (English questionnaire).**
(PDF)

## Acknowledgments

The authors would like to express heartfelt appreciation to all the faculty members and all the participants of the University for their active cooperation in this study.

## Author Contributions

**Conceptualization:** Aye Thazin Khine, Yu Mon Saw, Cho Thet Khaing, Htin Zaw Soe, Kyu Kyu Swe, Thu Nandar Saw, Nobuyuki Hamajima.

**Data curation:** Aye Thazin Khine, Cho Thet Khaing, Htin Zaw Soe, Thinzar Thike, Hein Htet, Thu Nandar Saw, Su Myat Cho, Nobuyuki Hamajima.

**Formal analysis:** Aye Thazin Khine, Yu Mon Saw, Zaw Ye Htut, Thinzar Thike, Hein Htet, Su Myat Cho.

**Investigation:** Aye Thazin Khine, Zaw Ye Htut, Cho Thet Khaing, Htin Zaw Soe, Kyu Kyu Swe, Thinzar Thike, Hein Htet, Thu Nandar Saw.

**Methodology:** Aye Thazin Khine, Yu Mon Saw, Zaw Ye Htut, Cho Thet Khaing, Htin Zaw Soe, Kyu Kyu Swe, Thu Nandar Saw, Su Myat Cho, Tetsuyoshi Kariya, Eiko Yamamoto, Nobuyuki Hamajima.

**Project administration:** Cho Thet Khaing, Kyu Kyu Swe, Thinzar Thike, Hein Htet, Thu Nandar Saw, Eiko Yamamoto.

**Resources:** Aye Thazin Khine, Cho Thet Khaing, Htin Zaw Soe, Nobuyuki Hamajima.

**Supervision:** Yu Mon Saw, Zaw Ye Htut, Cho Thet Khaing, Htin Zaw Soe, Kyu Kyu Swe, Thinzar Thike, Thu Nandar Saw, Tetsuyoshi Kariya, Eiko Yamamoto, Nobuyuki Hamajima.

**Validation:** Yu Mon Saw, Htin Zaw Soe, Thinzar Thike.

**Visualization:** Yu Mon Saw, Htin Zaw Soe.

**Writing – original draft:** Aye Thazin Khine, Zaw Ye Htut, Tetsuyoshi Kariya, Eiko Yamamoto, Nobuyuki Hamajima.

**Writing – review & editing:** Yu Mon Saw, Nobuyuki Hamajima.

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
