## [Decision Letter · Decision Letter 0]

10 Sep 2019

PONE-D-19-16609

Assessing risk factors and impact of cyberbullying victimization among university students in Myanmar: A cross-sectional study

PLOS ONE

Dear Dr. Saw,

Thank you for submitting your manuscript to PLOS ONE. After careful consideration, we feel that it has merit but does not fully meet PLOS ONE’s publication criteria as it currently stands. Therefore, we invite you to submit a revised version of the manuscript that addresses the points raised during the review process.

We would appreciate receiving your revised manuscript by Oct 25 2019 11:59PM. To enhance the reproducibility of your results, we recommend that if applicable you deposit your laboratory protocols in protocols.io, where a protocol can be assigned its own identifier (DOI) such that it can be cited independently in the future. For instructions see: http://journals.plos.org/plosone/s/submission-guidelines#loc-laboratory-protocols

We look forward to receiving your revised manuscript.

Kind regards,

Siyan Yi, MD, MHSc, PhD

Academic Editor

PLOS ONE

Journal Requirements:

2. Please include additional information regarding the survey or questionnaire used in the study and ensure that you have provided sufficient details that others could replicate the analyses. For instance, if you developed a questionnaire as part of this study and it is not under a copyright more restrictive than CC-BY, please include a copy, in both the original language and English, as Supporting Information

Additional Editor Comments (if provided):

Reviewers' comments:

Reviewer's Responses to Questions

**Comments to the Author**

1. Is the manuscript technically sound, and do the data support the conclusions?

Reviewer #1: Partly

Reviewer #2: Yes

Reviewer #3: Yes

2. Has the statistical analysis been performed appropriately and rigorously? 

Reviewer #1: N/A

Reviewer #2: I Don't Know

Reviewer #3: Yes

3. Have the authors made all data underlying the findings in their manuscript fully available?

Reviewer #1: Yes

Reviewer #2: No

Reviewer #3: Yes

4. Is the manuscript presented in an intelligible fashion and written in standard English?

Reviewer #1: Yes

Reviewer #2: Yes

Reviewer #3: Yes

5. Review Comments to the Author

Reviewer #1: Manuscript Number: PONE-D-19-16609

Article Type: Research Article

Full Title: Assessing risk factors and impact of cyberbullying victimization among university students in Myanmar: A cross-sectional study

Short Title: Risk factors

THE PROS

The paper is written with clarity, referencing recent theoretical and valid empirical findings from authors who are relevant in the proposed field and topic of research.

The paper has a very interesting title, even though the sample does not only consist of typically adolescent participants (students) aged 18-25, but more mature participants, as well (27-35), which implies different cognitive, emotional and behavioral traits, so the authors could consider adjusting the title so as to emphasize this sample age difference.

The structure of the paper is satisfactory, but some of the elements or contents should be moved to different sections of the paper (e.g. the research procedure description, sample characteristics), so as to better suit the study presentation.

The study design is valid, the sample size and characteristics are satisfactory, while the chosen title and research area of the paper present a scientifically important subject, due to the scarcity of research findings on this topic.

THE CONS (recommendations for improvement)

A) Page 4 – ...„To surf the worldwide information technology tide....“ – this section feels disconnected from the rest of the section content, so it would be recommended inserting latest data findings from a meta-analysis by Kowalski, R.M., Giumetti, G.W., Schroeder, A.N., Lattanner, M.R., 2014. Bullying in the digital age: a critical review and meta-analysis of cyberbullying research among youth. Psychol. Bull. 140, 1073–1137. This would help the authors better explain the importance of some of the studied socioeconomic characteristics.

B) Page 5 - Methods, Study area and participants - The authors should explicitly state the age status of the sample, because the sample used is very diversified age-wise, while age has been found to be a very significant factor when studying cyberbullying among the adolescent population.

C) The instrument's methodological properties or metric characteristics need more thorough clarification. Was the instrument used in previous similar studies, and what are its methodological properties (validity, reliability)? How many questions or scales and what kind of questions were used, and how was the intensity or frequency measured (Likert scale, semantic differential, scale type...)? This is recommended, because the authors state that “data were collected with a pre-tested, self-administered questionnaire. The questionnaire was developed by referring the domestic violence questionnaire from the 2015-2016 Myanmar Demographic and Health Survey [35], and the gender-based violence fact sheet published by the Department of Public Health, Ministry of Health and Sports, Myanmar [36].”

D) Page 6 – It would be recommended to omit the company and address of the company that produced SPSS, as it is not usual and unnecessary in the paper.

E) Page 7 - The second part of the section Ethical considerations contains information that is better suited for the Data collection section of the paper, as it describes the procedure of the instrument administration and implementation, so it should be moved there.

F) It would be recommended the authors use the term „Socioeconomic characteristics / status“ or „Socio-demographic characteristics / traits“ instead of using the term „Background characteristics“, as it does not describe the studied variables correctly.

G) Page 8 - instead of using „Magway“ as a point of reference or variable name, I would suggest using the „urban / rural“ variable distinction when presenting and discussing results (e.g., Table 1., Background data).

H) Table 1 should be revised so as to clearly show significant differences between the proposed variables (within the groups), as the significance is not visible in the table. For these kind of analyses, a t-test for independent samples (between 2 groups; e.g. marital status) or ANOVA (3 groups or more; e.g. age group) would be recommended, as chi-square tests are usually implemented with smaller samples containing less than 30 participants.

I) The authors should decide on the style of data presentation, as it is uncommon to describe the results via text, and then provide the same data visually / graphically in a table (e.g. Table 2, page 9). It should be mutually exclusive, so the authors should decide what style of data presentation to use and implement it uniformly throughout the paper.

J) The notes presented under Table 2 (page 10) related to the types of cyberbullying should be presented in the Introduction / Theoretical background of the paper, where such definitions of cyberbullying and related behavior should be referenced and clarified.

K) Table 3 is clear and presented well, but Table 4 needs revision, as such a style of presenting regression analysis results is unusual. Table 4 is confusing and complex, as the data is not validly and clearly presented. Regression analysis should be presented using APA standards, with percentages of variance explained, and clear indicators of R2 , F for change in R2, and R Square Change, as well as B, SE(B) and β coefficients as indicators of significant predictors. Maybe some examples from similar papers using the same type of statistical analysis would be helpful to the authors.

L) Discussion, page 14 - the authors reference studies on the relation of bullying and poor academic achievement, but it remains unclear whether the referenced studies report findings on traditional bullying or cyberbullying? ... „Many studies have demonstrated that among students, bullying is strongly correlated with poor academic performance or outcomes [24,31-33]. Students who suffered peer bullying received lower grades and, faced academic difficulties and/or worsened academic performance compared with their non-bullied peers [24,25,31-33].“

M) Considering the number of authors and their personal contribution to this paper, the Discussion section of the paper seems to be somewhat lacking, as it aims to describe only a segment of the results presented in the paper, while omitting study findings on various socio-economic characteristics that were established and presented in the authors' proposed study (risk behavior, suicide, use of substances, etc.). The authors should revise and further explain all their results, as socioeconomic variables (aside from gender or age) were found in recent studies to be significant in explaining cyberbullying perpetration and victimization.

Reviewer #2: 1. In starting of introduction, write ''smart phones with internet usage'' in place of ''mobile phone and internet usage''. Replace mobile phone with smart phone throughout text.

2. In second paragraph of introduction, revise in definition as “aggressive or intentional behavior causing harm, repeatedly and overtime, where it is difficult for the victim to defend himself or herself''.

3. Write as ''This cross-sectional study was conducted from August to September 2018 over a period of 2 months.''

4. Provide the self-administered questionnaire to reviewers.

5. In operational definition: How is stalking possible on Facebook? How a cybervictim can know he is being stalked?

6. Describe in detail about anonymization by a random identical number

7. Data was not collected for whatapps. Why? As you have collected it for Instagram, YouTube, Viber, WeChat and Twitter.

8. Study points out difficulty to concentrate following victimization in preceding 12 months. Is data collected about recent vs delayed cybervictimization on concentrating ability?

9. Conclusion should be made more concise.

10. A comparison group was needed to validate increased substance usage following cybervictimization.

11. Another form of cyberbullying is via different online apps and games. Please go through this correspondence published in Indian Pediatrics and if possible cite also.

https://indianpediatrics.net/dec2017/1056.pdf

12. Manuscript should be revised by a native English speaker.

Reviewer #3: The study examined cyberbullying among university students in Myanmar and analyzed the associations with adverse impacts on well-being, health and mental health, as well as academic performance of students.

Introduction:

The literature review is comprehensive in general. It is suggested to add a brief discussion on comparison between bullying and cyberbullying in paragraph two. Paragraph three: any explanation/characteristic on the prevalence difference among countries? Please amend throughout the manuscript on the "suffer cyberbullying" by clarifying "perpetration or victimization". Any comparison on the cyberbullying situations between Myanmar and other countries, e.g. western, developed countries? Please state clear the category of "student characteristic", and review previous findings on this variable when needed.

Method:

Please provide number of items, scoring method, example questions from each of the three sections.

Results:

It is suggested to rectify some expressions such as "adverse consequence" to "association", as cross-sectional studies do not provide causal inference to determine reason/consequence. The authors mentioned health and mental health correlates as one of the impacts to test in the Method section, it is suggested to present these variables in Table 3.

Discussion:

Some descriptions need citation, eg. paragraph 2 "evidence from previous studies is consistent with this study". Some descriptions need justification, eg. paragraph 2 "periodic screening...and counseling services...are also needed", or move it to the implication part for detailed discussion. The discussion on non-revealing of the cyberbullying experiences is comprehensive but kind of distracted the topic, please amend the logic. It is suggested to discuss why "no association between amount of time spent and cyberbullying was found in this study".

6. PLOS authors have the option to publish the peer review history of their article (what does this mean?). If published, this will include your full peer review and any attached files.

Reviewer #1: Yes: Goran Livazović

Reviewer #2: Yes: Shahid Akhtar Siddiqui

Reviewer #3: No

---

## [Author Response · Author response to Decision Letter 0]

22 Nov 2019

Response Letter

Ref: PONE-D-19-16609

Title: Assessing risk factors and impact of cyberbullying victimization among university students in Myanmar: A cross-sectional study

Academic Editor’s comment

Thank you for submitting your manuscript to PLOS ONE. After careful consideration, we feel that it has merit but does not fully meet PLOS ONE’s publication criteria as it currently stands. Therefore, we invite you to submit a revised version of the manuscript that addresses the points raised during the review process.

Authors’ response: Thank you very much for your valuable comments and suggestions that are very beneficial for revising and improving our manuscript. We have attempted to address all the comments of the reviewers and given the response for each and every comment. The revised and edited words and/or sentences are highlighted with yellow color in the manuscript. In the following response, we noted our response in blue color and reviewer’s comments in black color.

Reviewer #1

Q1: The paper is written with clarity, referencing recent theoretical and valid empirical findings from authors who are relevant in the proposed field and topic of research. The paper has a very interesting title, even though the sample does not only consist of typically adolescent participants (students) aged 18-25, but more mature participants, as well (27-35), which implies different cognitive, emotional and behavioral traits, so the authors could consider adjusting the title so as to emphasize this sample age difference. The structure of the paper is satisfactory, but some of the elements or contents should be moved to different sections of the paper (e.g. the research procedure description, sample characteristics), so as to better suit the study presentation. The study design is valid, the sample size and characteristics are satisfactory, while the chosen title and research area of the paper present a scientifically important subject, due to the scarcity of research findings on this topic.

Authors’ response: Thank you very much all reviewers for your valuable comments and suggestions. We have revised the manuscript according to your comments and suggestions.

Q2: A) Page 4 – ...„To surf the worldwide information technology tide....“ – this section feels disconnected from the rest of the section content, so it would be recommended inserting latest data findings from a meta-analysis by Kowalski, R.M., Giumetti, G.W., Schroeder, A.N., Lattanner, M.R., 2014. Bullying in the digital age: a critical review and meta-analysis of cyberbullying research among youth. Psychol. Bull. 140, 1073–1137. This would help the authors better explain the importance of some of the studied socioeconomic characteristics.

Authors’ response: Thank you so much for your comments. As suggested, we revised the previous paragraph in the introduction section: “To surf the worldwide information technology tide, widespread internet and mobile phone usage is inevitable in many countries. Today, mobile internet usage is ubiquitous among university students in Myanmar. University students who are in transitional phase of life into adulthood are willing to try and learn new things. Widespread mobile internet usage makes it easier for students to stay current, and even actively involved, in the things that interest them. With the use of this technology comes several pros and cons. Effective and efficient interventions are urgently needed to control undesirable aspects of the information technology tide such as cyberbullying. However, Myanmar being a developing country lags behind in the protection of technology consumers against cyberbullying. The very first cyber law in Myanmar is still in the early stages of development. Currently, protection against cyberbullying is indirectly provided under the Telecommunications Law and the Electronic Transactions Law.” to read as follows; “To surf the worldwide information technology tide, widespread internet and smart phone usage is inevitable in many countries making technology consumers more prone to cyberbullying. Although cyberbullying is common among middle school students [7,39], more than 30% of undergraduate students reported that they first experienced cyberbullying in college [40], and an equal victimization rate was found between male and female students [41]. Cyberbullying is very common in South East Asian (SEA) countries [42]. Studies related to cyberbullying in SEA countries observed 59.4% of cyberbullied victims among Facebook users in Singapore [43], 39.7% of young-adult (17 to 30 years) victims in Malaysia [44], 59% of cyberbullying victims in Thailand [45] and 80% of junior high school students experiencing cyberbullying victimization in Indonesia [46].

Today, smart phone internet usage is also ubiquitous among university students in Myanmar. University students who are in transitional phase of life into adulthood are willing to try and learn new things. Widespread smart phone internet usage makes it easier for students to stay current, and even actively involved, in the things that interest them. With the use of this technology comes several pros and cons. Effective and efficient interventions are urgently needed to control undesirable aspects of the information technology tide such as cyberbullying. However, Myanmar being a developing country lags behind in the protection of technology consumers against cyberbullying. The first cyber law in Myanmar is still in the early stages of development. Currently, protection against cyberbullying is indirectly provided under the Telecommunications Law and the Electronic Transactions Law.” [Introduction, Line 119 to 138, Page 5&6]

Q3: B) Page 5 - Methods, Study area and participants - The authors should explicitly state the age status of the sample, because the sample used is very diversified age-wise, while age has been found to be a very significant factor when studying cyberbullying among the adolescent population. 

Authors’ response: Thank you so much for your comments. As suggested, the sample used in this study is much diversified in age, while age has been found to be a very significant factor in cyberbullying studies. And the studied university has its own distinction in which it is the only university in Myanmar where the condensed health assistant course (CHA) for the public health staff under the Department of Public Health (DoPH) is located. The participants of CHA course started their career as Public Health Supervisor Grade 2 (PHS 2) in DoPH, Ministry of Health and Sports (MoHS), Myanmar. When they have at least 3 years of government service, they can sit for the promotion exam for Public Health Supervisor Grade 1 (PHS 1). After 3 years of service as PHS 1, they get another chance to sit for the entrance exam to join the CHA course. Therefore, the age of the students in this CHA course is much older than the students in regular 4-year health assistant course. For better understanding of this student characteristic, the previous paragraph for ‘Study area and participants’ under ‘Methods’ section; “The participants in this study were male and female university students aged 18 years and older, in their second to final year and those from the condensed health assistant course during the 2018-2019 academic year, at one medical university in Magway, Myanmar. The condensed health assistant course is different from the regular 4-year health assistant course. This 9-month condensed course is intended for the public health staff of the Department of Public Health, Ministry of Health and Sports, Myanmar. The participants who attended the lectures at the day of data collection and who gave their written informed consent to participate in the study were recruited. The total number of students attending the 2018 academic year at the university was 802. Among them, 453 students gave their consent to participate in this study. After data cleaning, 41 participants were excluded from the dataset due to missing data. Finally, 412 students (277 males and 135 females) were included in the data analysis.” is changed by adding a few more sentences describing the detail information about CHA course as follows; “The participants in this study were male and female university students aged 18 years and older in their second to final year, and from the condensed health assistant course during the 2018-2019 academic year, at a medical university in Magway, Myanmar. The condensed health assistant (CHA) course is different from the regular 4-year health assistant course. This 9-month condensed course is intended for the public health staff of the Department of Public Health (DoPH), Ministry of Health and Sports (MoHS), Myanmar. The participants of CHA course started their career as Public Health Supervisor Grade 2 (PHS 2) in DoPH, MoHS. When they have at least 3 years government service, they can sit for the promotion exam for Public Health Supervisor Grade 1 (PHS 1). After 3 years of service as PHS 1, they get another chance to sit for the entrance exam to join the CHA course. Therefore, the age of the students in the CHA course is much older than the students in regular 4-year health assistant course. The participants who attended the lectures at the day of data collection and who gave their written informed consent to participate in the study were recruited. The total number of students attending the 2018 academic year at the university was 802. Among them, 453 students gave their consent to participate in this study. After data cleaning, 41 participants were excluded from the dataset due to incomplete data. Finally, 412 students (277 males and 135 females) were included in the data analysis.” [Methods, Line 149 to 165, Page 6&7]

Q4: C) The instrument's methodological properties or metric characteristics need more thorough clarification. Was the instrument used in previous similar studies, and what are its methodological properties (validity, reliability)? How many questions or scales and what kind of questions were used, and how was the intensity or frequency measured (Likert scale, semantic differential, scale type...)? This is recommended, because the authors state that “data were collected with a pre-tested, self-administered questionnaire. The questionnaire was developed by referring the domestic violence questionnaire from the 2015-2016 Myanmar Demographic and Health Survey [35], and the gender-based violence fact sheet published by the Department of Public Health, Ministry of Health and Sports, Myanmar [36].” 

Authors’ response: Thank you so much for your comments. The self-administered questionnaire used in this study was developed by referring the domestic violence questionnaire from the 2015-2016 Myanmar Demographic and Health Survey [47], and the gender-based violence fact sheet published by the Department of Public Health, Ministry of Health and Sports, Myanmar [9]. The domestic violence questionnaire from the 2015-2016 Myanmar Demographic and Health Survey, and the statement about cyberbullying in the gender-based violence fact sheet were officially translated into Myanmar language version, and both of them were tested and validated by the Department of Public Health, Ministry of Health and Sports, Myanmar. The scales were not used in this self-administered questionnaire. The questionnaire consisted of three sections: 1) socio-demographic characteristics of the participants, 2) experiences with regard to cyberbullying victimization, and 3) association with adverse events following cyberbullying victimization. To detect socio-demographic characteristics of the participants, total eight questions were used. To detect experiences with regard to cyberbullying victimization, total seven questions were used. If the participant answered “Yes” to any of these seven questions _ Did anyone ever hack or stalk or use your Facebook/ social media account(s), or smart phone(s) without your consent and pretends to be you during the last 12 months? Did anyone ever use your picture online without your permission during the last 12 months? Did anyone ever tell lies or spread false rumors about you behind your back during the last 12 months? Did anyone ever send you humiliating/annoying/ mean texts or posts or sex chat during the last 12 months? Did anyone ever make upsetting phone calls or malicious prank calls to you during the last 12 months? Did anyone ever send unpleasant photos, sex pictures or videos to you against your will during the last 12 months? Did anyone ever take your photo(s), video(s) without your consent or humiliate you or threaten you during the last 12 months? _ he or she was categorized as the cyberbullying victim in this study. To detect adverse association of cyberbullying, total five questions were used. As suggested, the English and Myanmar version of the self-administered questionnaire that used in this study was submitted through the system (along with respond letter). [File name: English Questionnaires_Cyberbullying Final and Myanmar Questionnaires_Cyberbullying Final]

Q5: D) Page 6 – It would be recommended to omit the company and address of the company that produced SPSS, as it is not usual and unnecessary in the paper.

Authors’ response: Thank you so much for your comment. As suggested, we removed the company and address of the company that produced SPSS and revised the previous sentence “The data were analyzed using SPSS version 25 software (IBM SPSS Inc., Armonk, NY, USA).” into “The data were analyzed using SPSS version 25 software.” [Methods, Line 193, Page 8]

Q6: E) Page 7 - The second part of the section Ethical considerations contains information that is better suited for the Data collection section of the paper, as it describes the procedure of the instrument administration and implementation, so it should be moved there.

Authors’ response: Thank you so much for your comment. As suggested, we moved the second part of the section Ethical considerations, two sentences, “The aim of the study and questionnaire contents were explained to the participants prior to obtaining their consent. Data collection was conducted only after obtaining written informed consent from the participants.” to the data collection section. [Methods, Line 169 to 171, Page 7] And we removed the sentence “The aim of the study and questionnaire contents were explained to the participants prior to obtaining their consent.” from the Ethical considerations section. [Methods, Line 201 to 208, Page 8&9]

Q7: F) It would be recommended the authors use the term „Socioeconomic characteristics / status“ or „Socio-demographic characteristics / traits“ instead of using the term „Background characteristics“, as it does not describe the studied variables correctly.

Authors’ response: Thank you so much for your comments. As suggested, we changed the term “background characteristics” into “socio-demographic characteristics” in the whole manuscript. [Abstract, Line 33, Page 2; Introduction, Line 143, Page 6; Methods, Line 177 to 178, Page 7; Methods, Line 180 & 194, Page 8; Results, Line 212, Page 9; Table 1 title, Line 222, Page 9; Results, Line 264, Page 13; Discussion, Line 309, Page 16]

Q8: G) Page 8 - instead of using „Magway“ as a point of reference or variable name, I would suggest using the „urban / rural“ variable distinction when presenting and discussing results (e.g., Table 1., Background data).

Authors’ response: Thank you so much for your comments. Regarding the variable “Residence status of the participant”, we used “Magway and Other”. Myanmar is a multi-ethnic country comprised of eight major ethnic groups, sub-divided into 135 national races spoken over 100 languages throughout the country. In Myanmar, there are seven regions and seven states and one union territory, and each of them possess different socio-demographic situations. Students who want to become Health Assistant have to enroll only in this university as this is the only university available for Health Assistant courses in Myanmar. Therefore, students from different states and regions of Myanmar have to come and stay in Magway to attend this university. Majority of students (80.3%) in this study were not the locals in Magway who need to adapt the different socio-demographic situations during their stay in Magway, and we assumed this “Magway/ Other” variable as one of the important factors influencing cyberbullying events in this study. Therefore, we used the term “Magway/ Other” in this study.

Q9: H) Table 1 should be revised so as to clearly show significant differences between the proposed variables (within the groups), as the significance is not visible in the table. For these kind of analyses, a t-test for independent samples (between 2 groups; e.g. marital status) or ANOVA (3 groups or more; e.g. age group) would be recommended, as chi-square tests are usually implemented with smaller samples containing less than 30 participants.

Authors’ response: Thank you so much for your comments. In Table 1, we simply want to show the descriptive analysis results of the socio-demographic characteristics of the participants. Therefore, we would like to remove the p value column in Table 1. Please accept our sincere apology for any inconvenience that may occur. And the previous Table 1 was changed into as follows;

Table 1 Socio-demographic characteristics of the participants (N = 412)

Characteristics Total 

(N = 412) Males 

(N = 277) Females 

(N = 135) 

 N % N % N % 

Age group [completed years] (mean = 2.67, SD = 1.22) 

 18-22 319 77.4 236 85.2 83 61.5 

 23-27 63 15.3 29 10.5 34 25.2 

 28-35 30 7.3 12 4.3 18 13.3 

Marital status 

 Single 386 93.7 258 93.1 128 94.8 

 Married 26 6.3 19 6.9 7 5.2 

Total year(s) of study (mean = 2.67, SD = 1.22) 

 ≤1 87 21.1 35 12.6 52 38.5 

 2 113 27.4 90 32.5 23 17.0 

 3 79 19.2 55 19.9 24 17.8 

 ≥4 133 32.3 97 35.0 36 26.7 

Residence 

 Magway 81 19.7 61 22.0 20 14.8 

 Other 331 80.3 216 78.0 115 85.2 

Facebook usage 

 No 5 1.2 1 0.4 4 3.0 

 Yes 407 98.8 276 99.6 131 97.0 

Other social media usage§ 

 No 236 57.3 156 56.3 80 59.3 

 Yes 176 42.7 121 43.7 55 40.7 

Average hour(s) per day spent on social media (mean = 2.88, SD = 1.83) 

 ≤1 78 18.9 42 15.1 36 26.7 

 1 to 2 117 28.4 83 30.0 34 25.2 

 >2 217 52.7 152 54.9 65 48.1 

Witnessed psychological or physical or sexual violence or cyberbullying 

 No 55 13.3 45 16.2 10 7.4 

 Yes 357 86.7 232 83.8 125 92.6 

§Other social media usage: usage of Instagram or YouTube or Viber or WeChat or Bee Talk or Twitter [Results, Table 1, Line 222 to 223, Page 9&10]

Q10: I) The authors should decide on the style of data presentation, as it is uncommon to describe the results via text, and then provide the same data visually / graphically in a table (e.g. Table 2, page 9). It should be mutually exclusive, so the authors should decide what style of data presentation to use and implement it uniformly throughout the paper.

Authors’ response: Thank you so much for your comments. As suggested, in the ‘results’ section, we mainly used the tables to describe our data analysis results and reduced some text descriptions in the revised manuscript. Therefore, the previous text descriptions “Table 1 presents the background characteristics of the study participants according to sex. Among the 412 university students, 85.2% males and 61.5% females were aged between 18 and 22 years at the time of the survey (p<0.001). The majority of the participants (93.1% males and 94.8% females) were single. About one-third of the participants (35% males and 26.7% females) had been studying at the university for 4 years or more (p<0.001). It was observed that 78.0% of the males and 85.2% of the females came to Magway from different states and regions. Almost all the students (98.8%) used Facebook, and 42.7% of students also used other social media, such as Instagram, YouTube, Viber, WeChat and Twitter. Approximately half of the students (54.9% of males and 48.1% of females) spent more than two hours per day on social media (p = 0.020). Most of the students (83.8% of males and 92.6% of females) reported that they had ever witnessed psychological or physical or sexual violence or cyberbullying in their neighborhoods (p = 0.013).

Data on the participants’ experiences with regard to cyberbullying in the past 12 months are described in Table 2. Of all the participants, 44.2% (40.8% in males and 51.1% in females) had suffered cyberbullying in the past 12 months. Among the male students, 19.9% experienced someone using images of them online without their consent, 7.6% faced being told lies or having false rumors spread about them, and 5.1% experienced someone taking a photo/photos of them or videotaping them without their consent and using the photo(s) or video(s) to humiliate or threaten them. Among female students, 12.6% suffered being hacked, or stalked and experienced someone using their Facebook, other social media account(s) or mobile phone to impersonate them, 20% received humiliating, annoying or mean texts or posts or sex chats, 40.7% reported being disturbed by upsetting phone calls or malicious prank calls, and 17% received unpleasant photos, sexually explicit images or videos against their will. The majority of both male (77.9%) and female (65.2%) victims had experienced one or more of the types of cyberbullying mentioned above by age 20 or younger. Between male and female students, statistical differences were found in suffering cyberbullying in the past 12 months (p = 0.048), someone using images of them online without their consent (p = 0.016), receiving humiliating, annoying, or mean texts or posts or sex chats without their consent (p = 0.016), and receiving upsetting phone calls or malicious prank calls (p<0.001).

Table 3 describes the adverse consequences of cyberbullying among the victims. Of the male victims, 26.5% started or increased smoking, betel chewing (smokeless tobacco), or alcohol drinking, and 29.2% found it difficult to concentrate and understand lectures easily than usual after having suffered cyberbullying in the past 12 months. Among female victims, 7.2% considered attempting suicide after experiencing cyberbullying. The percentage of female students who had told anyone about their cyberbullying experience(s) was 60.9%, and the percentage of male students who had told anyone about their cyberbullying experience(s) was 45.1%. There was a statistical difference between males and females in starting or increasing smoking, betel chewing (smokeless tobacco), or alcohol drinking (p<0.001), difficulty concentrating and difficulty understanding lectures (p = 0.023), and telling anyone about the experience(s) (p = 0.039).

In the unadjusted analysis, having been cyberbullied was strongly associated with difficulty concentrating and difficulty understanding lectures (UOR = 6.81; 95% CI 3.31-13.98) and starting or increasing substance abuse (UOR = 4.57; 95% CI 2.25-9.31). In addition, cyberbullying victims were approximately three times more likely to tell anyone about their cyberbullying experience(s) (UOR = 3.58; 95% CI 2.34-5.47), and to have ever witnessed psychological or physical or sexual violence or cyberbullying in their neighborhoods (UOR = 3.26; 95% CI 1.66-6.38). Participants who were female (UOR = 1.52; 95% CI 1.00-2.30), older than 21 years of age (UOR = 1.60; 95% CI 1.02-2.51), or had studied at the university for three years or less (UOR = 1.98; 95% CI 1.29-3.05) were more likely to suffer cyberbullying (Table 4). 

Table 4 also shows the results of the multiple logistic regression analysis. In model I, background characteristics of the cyberbullying victims (i.e., age, sex, marital status, total year(s) of study, residence, Facebook usage, other social media (Instagram, YouTube, Viber, WeChat, Bee Talk, and Twitter) usage, average hour(s) per day spent on social media, and having ever witnessed psychological or physical or sexual violence or cyberbullying in the neighborhoods) were included. In model II, the adverse consequences of cyberbullying among the students (i.e., starting or increasing use of substances, experiencing difficulty concentrating and difficulty understanding lectures, considering attempting suicide, and telling anyone about the experience(s)) were added to the variables in model I. 

According to model I, students who had been studying at the university for three year or less (AOR = 1.81; 95% CI 1.14-2.85) and those who had ever witnessed psychological or physical or sexual violence or cyberbullying in their neighborhoods (AOR = 2.95; 95% CI 1.48-5.91) were more likely to suffer cyberbullying in the past 12 months.

In model II, cyberbullying victims were nearly four times more likely to face difficulties in concentrating and understanding the lectures compared with non-victims (AOR = 3.96; 95% CI 1.72-9.11). Students who told anyone about their experience(s) (AOR = 2.87; 95% CI 1.78-4.64) and who reported starting or increasing substance abuse (AOR = 2.37; 95% CI 1.02-5.49) were approximately two times more likely to be cyberbullying targets. Students who came to Magway from different states and regions to attend the university (AOR = 1.86; 95% CI 1.04-3.34) and those who were studying at the university for three years or less were more likely to have suffered cyberbullying in the past 12 months (Table 4).” were revised to read as follows; “Table 1 presents the socio-demographic characteristics of the participants according to sex. Among the 412 university students, majority (77.4%) of both male and female students were aged between 18 and 22 years at the time of the survey. About one-third of the participants (35% males and 26.7% females) had been studying at the university for four years or more. It was observed that 80.3% of the students came to Magway from different states and regions of Myanmar. Almost all the students (98.8%) used Facebook. Approximately half of the students (54.9% of males and 48.1% of females) spent more than two hours per day on social media. Most of the students (83.8% of males and 92.6% of females) reported that they had witnessed psychological or physical or sexual violence or cyberbullying in their neighborhoods.” [Results, Line 212 to 221, Page 9]

“Data on the participants’ experiences with regard to cyberbullying victimization in the past 12 months are described in Table 2. Of all the participants, 44.2% (40.8% in males and 51.1% in females) reported they suffered cyberbullying victimization in the past 12 months (p = 0.048). The majority of both male (77.9%) and female (65.2%) victims had experienced one or more of the types of cyberbullying victimization mentioned above by age 20 or younger.” [Results, Line 225 to 230, Page 10]

“Table 3 describes the adverse association of cyberbullying victimization among the victims. Of the male victims, 26.5% started or increased smoking, betel chewing (smokeless tobacco), or alcohol drinking (p<0.001), and 29.2% found it more difficult to concentrate and understand lectures than usual after having suffered cyberbullying victimization in the past 12 months (p = 0.023). Among female victims, 7.2% considered attempting suicide after experiencing cyberbullying victimization.” [Results, Line 239 to 244, Page 11]

“In the unadjusted analysis, having been cyberbullied was strongly associated with difficulty in concentrating and understanding lectures (UOR = 6.81; 95% CI 3.31-13.98) and starting or increasing substance abuse (UOR = 4.57; 95% CI 2.25-9.31). In addition, cyberbullying victims were approximately three times more likely to tell anyone about their experience(s) (UOR = 3.58; 95% CI 2.34-5.47), and to have ever witnessed psychological, physical or sexual violence, or cyberbullying in their neighborhoods (UOR = 3.26; 95% CI 1.66-6.38). Participants who were female (UOR = 1.52; 95% CI 1.00-2.30), older than 21 years of age (UOR = 1.60; 95% CI 1.02-2.51), or had studied at the university for three years or less (UOR = 1.98; 95% CI 1.29-3.05) were more likely to suffer cyberbullying victimization (Table 4).” [Results, Line 249 to 258, Page 12]

“Tables 5 and 6 show the results of the multiple logistic regression analysis. In model I, socio-demographic characteristics of the cyberbullying victims ¬_ age, sex, marital status, total year(s) of study, residence, Facebook usage, other social media (Instagram, YouTube, Viber, WeChat, Bee Talk, and Twitter) usage, average hour(s) per day spent on social media, and having ever witnessed psychological or physical or sexual violence or cyberbullying in the neighborhoods _ were included (Table 5). In model II, the adverse events following cyberbullying victimization among the students _ starting or increasing use of substances, experiencing difficulty in concentrating and understanding lectures, considering attempting suicide, and telling anyone about the cyberbullying victimization experience(s) _ were added to the variables in model I (Table 6).” [Results, Line 264 to 273, Page 13]

“According to model I, students who had been studying at the university for three years or less (AOR = 1.81; 95% CI 1.14-2.85) and those who had witnessed psychological or physical or sexual violence or cyberbullying in their neighborhoods (AOR = 2.95; 95% CI 1.48-5.91) were more likely to suffer cyberbullying victimization in the past 12 months (Table 5).” [Results, Line 274 to 278, Page 13 & 14]

“In model II, cyberbullying victims were nearly four times more likely to face difficulties in concentrating and understanding lectures, in comparison to non-victims (AOR = 3.96; 95% CI 1.72-9.11). Students who told anyone about their experience(s) (AOR = 2.87; 95% CI 1.78-4.64) and who reported starting or increasing substance abuse (AOR = 2.37; 95% CI 1.02-5.49) were approximately two times more likely to be cyberbullying targets. Students who came to Magway from different states and regions to attend the university (AOR = 1.86; 95% CI 1.04-3.34) and those who were studying at the university for three years or less were more likely to have suffered cyberbullying victimization in the past 12 months (Table 6).” [Results, Line 2856 to 294, Page 14 & 15]

Q11: J) The notes presented under Table 2 (page 10) related to the types of cyberbullying should be presented in the Introduction / Theoretical background of the paper, where such definitions of cyberbullying and related behavior should be referenced and clarified.

Authors’ response: Thank you so much for your comments. As suggested, we moved the notes presented under Table 2 (Page 10 in the previous manuscript) related to the types of cyberbullying to the introduction section (paragraph 3 in the revised manuscript) as follows; “There are many different ways to perpetrate cyberbullying through internet or smart phone technology, which at its most extreme includes online suicide challenge games [8]. These include hacking or stalking a person’s Facebook or social media account(s) or smart phone, and impersonating them, using a person’s picture online without his or her consent, telling lies or spreading false rumors about a person behind his or her back, sending humiliating, annoying or mean texts or posts or sex chat, making upsetting phone calls or malicious prank calls, sending unpleasant photos, sexually explicit images or videos to a person without their consent, and taking a photo/photos of a person or videotaping a person without their consent and using the photo(s) or video(s) to humiliate or threaten them [6,9,10]. ” [Introduction; Line 78 to 87, Page 3&4]

Q12: K) Table 3 is clear and presented well, but Table 4 needs revision, as such a style of presenting regression analysis results is unusual. Table 4 is confusing and complex, as the data is not validly and clearly presented. Regression analysis should be presented using APA standards, with percentages of variance explained, and clear indicators of R2, F for change in R2, and R Square Change, as well as B, SE(B) and β coefficients as indicators of significant predictors. Maybe some examples from similar papers using the same type of statistical analysis would be helpful to the authors.

Authors’ response: Thank you so much for your comments. As suggested, we separated the regression analysis results of the previous Table 4 into three tables “Table 4, Table 5 and Table 6 using APA standards” as follows: “Table 4 Unadjusted analysis of maximum likelihood and odds ratio estimates for cyberbullying victimization among students in the past 12 months (N = 412)

 Maximum likelihood estimates OR estimates

Parameter df Estimate(B) SE Wald χ2 P value Parame-ter UOR 95% CI

Age [completed years] (mean = 2.67, SD = 1.22)

>21 1 0.47 0.23 4.19 0.041 vs. ≤21 1.60 1.02-2.51

Sex 

Female 1 0.42 0.21 3.90 0.048 vs. Male 1.52 1.00-2.30

Marital status 

Married 1 0.42 0.41 1.04 0.308 vs. Single 1.51 0.69-3.36

Total year(s) of study (mean = 2.67, SD = 1.22)

≤3 1 0.68 0.22 9.66 0.002 vs. >3 1.99 1.29-3.05

Residence 

Other 1 0.37 0.26 2.07 0.150 vs. Magway 1.44 0.88-2.38

Facebook usage

Yes 1 0.17 0.92 0.04 0.850 vs. No 1.19 0.20-7.20

Other social media usage§

Yes 1 0.37 0.20 3.43 0.064 vs. No 1.45 0.98-2.15

Average hour(s) per day spent on social media (mean = 2.88, SD = 1.83)

1 to 2 1 -0.40 0.30 1.83 0.176 vs. ≤1 0.67 0.38-1.20

>2 1 -0.23 0.27 0.77 0.382 vs. ≤1 0.79 0.47-1.33

Witnessed psychological or physical or sexual violence or cyberbullying

Yes 1 1.18 0.34 11.84 0.001 vs. No 3.26 1.66-6.38

Started or increased use of substance¶

Yes 1 1.52 0.36 17.56 <0.001 vs. No 4.58 2.25-9.31

Being difficult to concentrate and understand lectures

Yes 1 1.92 0.37 27.24 <0.001 vs. No 6.81 3.31-13.98

Considered attempting suicide in the past 12 months

Yes 1 2.69 1.05 6.57 0.010 vs. No 14.73 1.88-115.20

Told anyone about cyberbullying victimization experience(s)

Yes 1 1.28 0.22 34.68 <0.001 vs. No 3.58 2.34-5.47

§Other social media usage: usage of Instagram or YouTube or Viber or WeChat or Bee Talk or Twitter 

¶substance: smoking or betel chewing (smokeless tobacco) or alcohol drinking” [Results: Line 259 to 262, Page 12&13]

“Table 5 Adjusted analysis of maximum likelihood and odds ratio estimates for cyberbullying victimization among students in the past 12 months (N = 412) [Model I✡]

 Maximum likelihood estimates OR estimates

Parameter df Estimate(B) SE Wald χ2 P value Parame-ter AOR 95% CI

Age [completed years] (mean = 2.67, SD = 1.22)

>21 1 0.18 0.28 0.44 0.510 vs. ≤21 1.20 0.70-2.06

Sex

Female 1 0.25 0.23 1.21 0.270 vs. Male 1.29 0.82-2.02

Marital status

Married 1 0.15 0.45 0.11 0.740 vs. Single 1.16 0.48-2.83

Total year(s) of study (mean = 2.67, SD = 1.22)

≤3 1 0.59 0.23 6.42 0.011 vs. >3 1.81 1.14-2.85

Residence 

Other 1 0.33 0.27 1.52 0.218 vs. Magway 1.39 0.82-2.34

Facebook usage

Yes 1 0.62 0.95 0.43 0.514 vs. No 1.86 0.29-11.91

Other social media usage§

Yes 1 0.19 0.22 0.73 0.392 vs. No 1.21 0.79-1.86

Average hour(s) per day spent on social media (mean = 2.88, SD = 1.83)

1 to 2 1 -0.27 0.31 0.74 0.391 vs. ≤1 0.77 0.41-1.41

>2 1 -0.30 0.29 0.01 0.918 vs. ≤1 0.97 0.55-1.70

Witnessed psychological or physical or sexual violence or cyberbullying

Yes 1 1.08 0.35 9.35 0.002 vs. No 2.95 1.48-5.91

§Other social media usage: usage of Instagram or YouTube or Viber or WeChat or Bee Talk or Twitter 

✡ Model I adjusted for age, sex, marital status, total year(s) of study, residence, Facebook usage, other social media usage, average hour(s) per day spent on social media, and witnessing psychological or physical or sexual violence or cyberbullying in their neighborhoods.” [Results: Line 279 to 284, Page 14]

“Table 6 Adjusted analysis of maximum likelihood and odds ratio estimates for cyberbullying victimization among students in the past 12 months (N = 412) [Model II✲]

 Maximum likelihood estimates OR estimates

Parameter df Estimate(B) SE Wald χ2 P value Parameter AOR 95% CI

Age [completed years] (mean = 2.67, SD = 1.22)

>21 1 0.11 0.30 0.14 0.705 vs. ≤21 1.12 0.62-2.03

Sex 

Female 1 0.28 0.26 1.20 0.272 vs. Male 1.33 0.80-2.19

Marital status

Married 1 0.59 0.47 1.56 0.211 vs. Single 1.81 0.72-4.56

Total year(s) of study (mean = 2.67, SD = 1.22)

≤3 1 0.52 0.25 4.29 0.038 vs. >3 1.69 1.03-2.77

Residence 

Other 1 0.62 0.30 4.30 0.038 vs. Magway 1.86 1.04-3.34

Facebook usage

Yes 1 1.08 1.01 1.13 0.289 vs. No 2.93 0.40-21.32

Other social media usage§

Yes 1 0.28 0.24 1.34 0.248 vs. No 1.32 0.83-2.10

Average hour(s) per day spent on social media (mean = 2.88, SD = 1.83)

1 to 2 1 -0.11 0.34 0.11 0.743 vs. ≤1 0.89 0.46-1.74

>2 1 0.15 0.31 0.23 0.634 vs. ≤1 1.16 0.63-2.14

Witnessed psychological or physical or sexual violence or cyberbullying

Yes 1 0.66 0.37 3.13 0.077 vs. No 1.93 0.93-4.00

Started or increased use of substance¶

Yes 1 0.86 0.43 4.04 0.044 vs. No 2.37 1.02-5.49

Being difficult to concentrate and understand lectures

Yes 1 1.38 0.43 10.45 0.001 vs. No 3.96 1.72-9.11

Considered attempting suicide in the past 12 months

Yes 1 1.32 1.10 1.46 0.227 vs. No 3.76 0.44-32.22

Told anyone about cyberbullying victimization experience (s)

Yes 1 1.05 0.25 18.52 <0.001 vs. No 2.87 1.78-4.64

§Other social media usage: usage of Instagram or YouTube or Viber or WeChat or Bee Talk or Twitter 

§Other social media usage: usage of Instagram or YouTube or Viber or WeChat or Bee Talk or Twitter 

¶substance: smoking or betel chewing (smokeless tobacco) or alcohol drinking

✲Model II adjusted for age, sex, marital status, total year(s) of study, residence, Facebook usage, other social media (Instagram or YouTube or Viber or WeChat or Bee Talk or Twitter) usage, average hour(s) per day spent on social media, ever witnessing psychological or physical or sexual violence or cyberbullying in the neighborhoods, started or increased use of substance, being difficult to concentrate and understand lectures, considering attempting suicide in the past 12 months, and telling anyone about cyberbullying experience(s).” [Results Line 295 to 304, Page 15]

Q13: L) Discussion, page 14 - the authors reference studies on the relation of bullying and poor academic achievement, but it remains unclear whether the referenced studies report findings on traditional bullying or cyberbullying? ... „Many studies have demonstrated that among students, bullying is strongly correlated with poor academic performance or outcomes [24,31-33]. Students who suffered peer bullying received lower grades and, faced academic difficulties and/or worsened academic performance compared with their non-bullied peers [24,25,31-33].“

Authors’ response: Thank you so much for your comments. “Bullying” in these references [24, 25, 31-33] means “traditional bullying”. For better understanding, we revised the previous discussion section “Many studies have demonstrated that among students, bullying is strongly correlated with poor academic performance or outcomes [24,31-33]. Students who suffered peer bullying received lower grades and, faced academic difficulties and/or worsened academic performance compared with their non-bullied peers [24,25,31-33].” into to read as follows; “Many studies have demonstrated that among students, traditional bullying as well as cyberbullying is strongly related to poor academic performance or outcomes [24-26,28,35-37]. Students who suffered peer bullying received lower grades, and faced academic difficulties and/or worsened academic performance compared with their non-bullied peers [24-26,28,29,35-37]. In this context, this study is consistent with the evidence from previous studies [24-26,28,29,35-37].” [Discussion, Line 322 to 328, Page 16]

Q14: M) Considering the number of authors and their personal contribution to this paper, the Discussion section of the paper seems to be somewhat lacking, as it aims to describe only a segment of the results presented in the paper, while omitting study findings on various socio-economic characteristics that were established and presented in the authors' proposed study (risk behavior, suicide, use of substances, etc.). The authors should revise and further explain all their results, as socioeconomic variables (aside from gender or age) were found in recent studies to be significant in explaining cyberbullying perpetration and victimization.

Authors’ response: Thank you so much for your comments. As suggested, we revised the previous discussion section by adding several sentences to cover all the socio-demographic variables which are significant into to read as follows; 

“In the unadjusted analysis, being female and being older than 21 years of age were associated with increased risk of being cyberbullying victims. Moreover, cyberbullying victimization is associated with having suicidal ideas in the past 12 months.” [Discussion, Line 318 to 321, Page 16], and “In addition, suicidal ideation was positively associated with cyberbullying victimization in the past 12 months in this study which supported a similar association found in the studies conducted among cyberbullied victims in the United States, France, Hong Kong and several other countries [33,34,52,53]. These studies also observed that cyberbullying victimization is much more likely to be positively associated with suicidal ideation compared to traditional bullying or cyberbullying perpetration alone [33,34,52,53].” [Discussion, Line 385 to 391, Page 19]

Reviewer #2

Q1: In starting of introduction, write ''smart phones with internet usage'' in place of ''mobile phone and internet usage''. Replace mobile phone with smart phone throughout text.

Authors’ response: Thank you so much for your comment. As suggested, we replaced the words “mobile phone and internet usage” into “smart phones with internet usage,” as well as “mobile phone” into “smart phone” throughout the manuscript. [Introduction: Line 59, Page 3; Line 119 to 120, Page 5; Line 129, Page 5; Line 131, page 5], [Methods, Line 185 & 186, Page 8], and [Results, Line 234, Page 11].

Q2: In second paragraph of introduction, revise in definition as “aggressive or intentional behavior causing harm, repeatedly and overtime, where it is difficult for the victim to defend himself or herself''.

Authors’ response: Thank you so much for your comment. As suggested, we paraphrased the definition “aggressive or intentional behavior causing harm, repeatedly and overtime, where it is difficult for the victim to defend himself or herself'' into “Cyberbullying characteristics are similar to traditional bullying, which is an act or behavior which is intentional, aggressive and repetitive in nature, causing harm to a victim such that the victim finds it difficult to defend themselves owing to power imbalance [5].” [Introduction, Line 70 to 72, Page 3]

Q3: Write as ''This cross-sectional study was conducted from August to September 2018 over a period of 2 months.''

Authors’ response: Thank you so much for your comment. As suggested, we revised the previous sentence “This cross-sectional study was conducted between August, 2018, and September, 2018.” into to read as follows; “This cross-sectional study was conducted from August to September 2018 over a period of two months.'' [Methods, Line 167 to 168, Page 7]

Q4: Provide the self-administered questionnaire to reviewers.

Authors’ response: Thank you so much for your comment. As suggested, both English and Myanmar version of the self-administered questionnaire used in this study was provided to the reviewers in separate files. [File name: English Questionnaires_Cyberbullying Final and Myanmar Questionnaires_Cyberbullying Final]

Q5: In operational definition: How is stalking possible on Facebook? How a cybervictim can know he is being stalked?

Authors’ response: Thank you so much for your comment. Stalking can occur on social media such as Facebook in which the perpetrator uses fake accounts and he or she is always monitoring the victim’s profile, posts, daily routine, comments, or the victim’s whereabouts or communication via the chat box(es). A cyber victim can know he or she is being stalked when the stalker’s action(s) appears not only on Facebook but also outside social media, disturbing his or her daily life and making the victim feel unsafe.

Q6: Describe in detail about anonymization by a random identical number.

Authors’ response: Thank you so much for your comment. At first, we’d like to request to allow us to correct the spelling ‘a random identical number’ into ‘a random identity number’. [Method, Line 207, Page 9]. Anonymization by a random identity number in this study means at first, each of the self-administered questionnaire was given a serial random number before data collection. After getting the written informed consent from the participants, these self-administered questionnaires were distributed to each of the class according to the total number of voluntary participants in these classes to calculate the actual response rate. After recollecting all the distributed questionnaires from each class, the incomplete answers were removed and all the complete answer sheets were recoded with the new identity number before inputting these data in SPSS.

Q7: Data was not collected for whatapps. Why? As you have collected it for Instagram, YouTube, Viber, WeChat and Twitter.

Authors’ response: Thank you so much for your comment. For the question “Name social media you use most commonly?”, the multiple-choice answers for commonly used social media in Myanmar were given such as Facebook, Instagram, YouTube, Viber, We Chat and Bee Talk. In addition to these common social media, a blank space is opened to write down any other social media the respondent might use commonly in the answer section. WhatsApp is not so popular in Myanmar and no one answered using WhatsApp as their commonly used social media in this study. Therefore, the data about WhatsApp is not included in this study.

Q8: Study points out difficulty to concentrate following victimization in preceding 12 months. Is data collected about recent vs delayed cybervictimization on concentrating ability?

Authors’ response: Thank you so much for your comment. This cross-sectional study used self-administered questionnaire and so the data collected regarding difficulty to concentrate following victimization in preceding 12 months were self-reported data from the participants. And, we didn’t use any measurement scale to detect about difficulty to concentrate and understand the lectures following victimization in preceding 12 months in this study. So, recent vs delayed cybervictimization on students’ concentrating ability cannot be differentiated and this fact is described as one of the limitations of this study. Therefore, we revised the previous sentence in the discussion section regarding study limitations “The extent of the adverse consequences of cyberbullying could not be measured accurately in this study.” into to read as follows; “The extent of the adverse association of cyberbullying could not be measured accurately as no measurement scale or a comparison group was used in this study.” [Discussion, Line 403 to 405, Page 19]

Q9: Conclusion should be made more concise.

Authors’ response: Thank you so much for your comment. As suggested, we revised the previous conclusion “To be concluded, two out of five students suffered cyberbullying in the past 12 months and only half of the victims told anyone about this experience(s). Having been cyberbullied was found to be positively associated with difficulty concentrating and difficulty understanding lectures, and increasing or starting substance abuse. Students who had ever witnessed psychological or physical or sexual violence or cyberbullying in their neighborhoods were more likely to suffer cyberbullying than their peers. Non-resident students and students who had been studying in the university for three years or less were at higher risk of suffering cyberbullying. Being female and being older than 21 years of age were positively associated with having been cyberbullied in the past 12 months. Awareness raising campaigns and educational programs for cyber safety are recommended. Periodic screening for cyberbullying, as well as counselling services and counter measures to prevent and mitigate the adverse consequences of cyberbullying are urgently needed among university students in Myanmar.” into to read as follows; “To conclude, two out of five students suffered cyberbullying victimization in the past 12 months. Non-resident students and students who had been studying in the university for three years or less were at a higher risk of being cyberbullying victims. Students who had witnessed psychological, physical or sexual violence, or cyberbullying in their neighborhoods were more likely to experience cyberbullying victimization. Cyberbullying victims were found to be positively associated with difficulty in concentrating and understanding lectures, and starting or increasing substance abuse. Periodic screening for cyberbullying, counseling services, cyber-safety educational programs, and awareness raising campaigns are urgently needed for university students in Myanmar.” [Conclusion, Line 420 to 428, Page 20]

 Q10: A comparison group was needed to validate increased substance usage following cybervictimization.

Authors’ response: Thank you so much for your comment. This is a cross-sectional study and therefore, a comparison group could not be used to validate increased substance usage following cybervictimization and we described this fact as one of the study limitations by revising the previous sentence “The extent of the adverse consequences of cyberbullying could not be measured accurately in this study.” into to read as follows; “The extent of the adverse association of cyberbullying could not be measured accurately as no measurement scale or a comparison group was used in this study.” [Discussion, Line 403 to 405, Page 19]

Q11: Another form of cyberbullying is via different online apps and games. Please go through this correspondence published in Indian Pediatrics and if possible cite also.

https://indianpediatrics.net/dec2017/1056.pdf

Authors’ response: Thank you so much for your comment. As suggested, we newly added another form of cyberbullying via online games in introduction section as follows; “There are many different ways to perpetrate cyberbullying through internet or smart phone technology, which at its most extreme includes online suicide challenge games [8].” [Introduction, Line 78 to 79, Page 3]

Q12: Manuscript should be revised by a native English speaker.

Authors’ response: Thank you so much for your comment. As suggested, the revised manuscript was checked and corrected by a native English speaker who is an expert in public health field.

Reviewer #3

I. Introduction:

Q1: The literature review is comprehensive in general. It is suggested to add a brief discussion on comparison between bullying and cyberbullying in paragraph two. 

Authors’ response: Thank you so much for your comment. As suggested, we added a brief discussion on comparison between bullying and cyberbullying in the paragraph two of the previous manuscript “Cyberbullying can be regarded as “a new violence type of the era” [1] especially among school children, adolescents, and youths [2-4]. It is also considered a hidden epidemic. It can have numerous adverse effects on people from all walks of life, and due to the technology involved, it can occur at any time [3-5]. In general, bullying can be described as “aggressive or intentional behavior causing harm, repeatedly and overtime, where it is difficult for the victim to defend him or herself” [5]. Cyberbullying is regarded as “an aggressive and intentional act that is carried out using electronic forms of contact by a group or an individual repeatedly and over time against a victim who cannot easily defend him or herself” [6]” into to read as follows; “Cyberbullying can be regarded as “a new violence type of the era” [1] especially among school children, adolescents, and youths [2-4]. It is also considered a hidden epidemic. It can have numerous adverse effects on people from all walks of life, and owing to the involvement of technology, it can occur any time [3-5]. Cyberbullying is defined as “an aggressive and intentional act that is carried out using electronic forms of contact by a group or an individual repeatedly and over time against a victim who cannot easily defend him or herself” [6]. Cyberbullying characteristics are similar to traditional bullying, which is an act or behavior which is intentional, aggressive and repetitive in nature, causing harm to a victim such that the victim finds it difficult to defend themselves owing to power imbalance [5]. However, in cyberbullying, the perpetrators can stay anonymous allowing greater potential to do or say more harmful things to the victims than they would in personal relations [7]. Moreover, the perpetrators can reach out to the victim 24 hours a day via internet accessibility and can send annoying mails, messages or spread rumors online which can draw a larger audience than traditional bullying [7].” [Introduction, Line 64 to 77, Page 3]

Q2: Paragraph three: any explanation/characteristic on the prevalence difference among countries?

Authors’ response: Thank you so much for your comments. As suggested, we added the explanation on the prevalence difference among countries in the previous paragraph “Numerous studies have been carried out to assess the status of cyberbullying among middle and high school students; however, few have focused on university students [4-14]. Several studies conducted among undergraduates on the prevalence of cyberbullying have found that the prevalence of cyberbullying ranges from 10% to 60%, differing among countries [1,7-9,13,14]. Astonishingly, in most studies, the victim reporting rate was observed to be very low compared with the prevalence rate [6,11,15-19].” into to read as follows; “Numerous studies have been conducted to assess the status of cyberbullying among middle and high school students; however, only few have focused on university students [4-6,11-18]. Several studies conducted among undergraduates on the prevalence of cyberbullying have found that it ranges from 10% to 60%, across different countries [1,11-13,17,18]. The differing prevalence of cyberbullying may be due to the different operational definitions for cyberbullying, different methodologies such as classification depending on frequencies (at least once or several times, etc.) or reported time interval (within one week, one month, one year, or lifetime, etc.), and different socio-demographic characteristics of the study participants in various studies [7]. Astonishingly, in most studies, the rate of reporting by victims was observed to be very low compared with the prevalence of cyberbullying [6,15,19-23].” [Introduction, Line 88 to 98, Page 4]

Q3: Please amend throughout the manuscript on the "suffer cyberbullying" by clarifying "perpetration or victimization"

Authors’ response: Thank you so much for your comment. As suggested, we clarified “suffer cyberbullying” into “cyberbullying victimization or cyberbullying victims or to be victimized” throughout the manuscript as this study mainly focused on the cyberbullying victimization experience(s) of the participants. [Abstract, Line 32, 34, 42, 45, 49, 50, 52 & 54, Page 2], [Introduction, Line 99, 103, 105 to 106, 109, 112, 116, 142 & 144, Page 3 to 6], [Methods, Line 178 to179, 180 to 181, 181 to 182 & 195, Page 6 to 8], [Results, Line 225, 227, 229, 231 to 233 (Table 2), 239, 242, 244 & 245 to 246 (Table 3), 257 to 258, 260 to 261 (Table 4), 277, 280 (Table 5 title), 293, 296 to 303 (Table 6), Page 10 to 15], [Discussion, Line 308, 310, 311, 316, 318, 320, 359, 366, 369, 371, 375, 379, 386, 387, 389, 394, 395 to 396, & 402, Page 15 to 19] and [Conclusion, Line 420, 422, & 424, Page 20]

Q4: Any comparison on the cyberbullying situations between Myanmar and other countries, e.g. western, developed countries?

Authors’ response: Thank you so much for your comments. This study is the first to conduct about cyberbullying situation among university students in Myanmar, and to emphasize this, the previous 7th paragraph of the introduction section “Ambiguous protection under the two laws leaves the student population at risk. Therefore, this study aimed to determine the percentage of university students who suffered cyberbullying within the past 12 months, and the associations between students’ background characteristics, adverse consequences and cyberbullying.” is changed into 9th paragraph in the introduction section to read as follows; “These laws only provide an ambiguous protection, leaving the student population at risk. There is a lack of statistical data assessing current cyberbullying situation among university students in Myanmar [42]. Therefore, this study aimed to determine the percentage of university students who suffered cyberbullying victimization in the past 12 months, and the associations between students’ socio-demographic characteristics, adverse events following cyberbullying and victimization.” [Introduction, Line 139 to 144, Page 6]

Cyberbullying situation among Southeast Asian countries and other countries were added in the previous 6th paragraph of introduction section “To surf the worldwide information technology tide, widespread internet and mobile phone usage is inevitable in many countries. Today, mobile internet usage is ubiquitous among university students in Myanmar. University students who are in transitional phase of life into adulthood are willing to try and learn new things. Widespread mobile internet usage makes it easier for students to stay current, and even actively involved, in the things that interest them. With the use of this technology comes several pros and cons. Effective and efficient interventions are urgently needed to control undesirable aspects of the information technology tide such as cyberbullying. However, Myanmar being a developing country lags behind in the protection of technology consumers against cyberbullying. The very first cyber law in Myanmar is still in the early stages of development. Currently, protection against cyberbullying is indirectly provided under the Telecommunications Law and the Electronic Transactions Law.” into the paragraph 7th and 8th in the introduction section of the revised manuscript to read as follows; “To surf the worldwide information technology tide, widespread internet and smart phone usage is inevitable in many countries making technology consumers more prone to cyberbullying. Although cyberbullying is common among middle school students [7,39], more than 30% of undergraduate students reported that they first experienced cyberbullying in college [40], and an equal victimization rate was found between male and female students [41]. Cyberbullying is very common in South East Asian (SEA) countries [42]. Studies related to cyberbullying in SEA countries observed 59.4% of cyberbullied victims among Facebook users in Singapore [43], 39.7% of young-adult (17 to 30 years) victims in Malaysia [44], 59% of cyberbullying victims in Thailand [45] and 80% of junior high school students experiencing cyberbullying victimization in Indonesia [46].

Today, smart phone internet usage is also ubiquitous among university students in Myanmar. University students who are in transitional phase of life into adulthood are willing to try and learn new things. Widespread smart phone internet usage makes it easier for students to stay current, and even actively involved, in the things that interest them. With the use of this technology comes several pros and cons. Effective and efficient interventions are urgently needed to control undesirable aspects of the information technology tide such as cyberbullying. However, Myanmar being a developing country lags behind in the protection of technology consumers against cyberbullying. The first cyber law in Myanmar is still in the early stages of development. Currently, protection against cyberbullying is indirectly provided under the Telecommunications Law and the Electronic Transactions Law.” [Introduction, Line 119 to 138, Page 5&6]

Q5: Please state clear the category of "student characteristic", and review previous findings on this variable when needed.

Authors’ response: Thank you so much for your comments. As suggested, the sample used in this study is much diversified in age, while age has been found to be a very significant factor in cyberbullying studies. And the studied university has its own distinction in which it is the only university in Myanmar where the condensed health assistant course (CHA) for the public health staff under the Department of Public Health (DoPH) is located. The participants of CHA course started their career as Public Health Supervisor Grade 2 (PHS 2) in DoPH, Ministry of Health and Sports (MoHS), Myanmar. When they have at least 3 years of government service, they can sit for the promotion exam for Public Health Supervisor Grade 1 (PHS 1). After 3 years of service as PHS 1, they get another chance to sit for the entrance exam to join the CHA course. Therefore, the age of the students in this CHA course is much older than the students in regular 4-year health assistant course. For better understanding of this student characteristic, the previous paragraph for ‘Study area and participants’ under ‘Methods’ section; “The participants in this study were male and female university students aged 18 years and older, in their second to final year and those from the condensed health assistant course during the 2018-2019 academic year, at one medical university in Magway, Myanmar. The condensed health assistant course is different from the regular 4-year health assistant course. This 9-month condensed course is intended for the public health staff of the Department of Public Health, Ministry of Health and Sports, Myanmar. The participants who attended the lectures at the day of data collection and who gave their written informed consent to participate in the study were recruited. The total number of students attending the 2018 academic year at the university was 802. Among them, 453 students gave their consent to participate in this study. After data cleaning, 41 participants were excluded from the dataset due to missing data. Finally, 412 students (277 males and 135 females) were included in the data analysis.” is changed by adding a few more sentences describing the detail information about CHA course as follows; The participants in this study were male and female university students aged 18 years and older in their second to final year, and from the condensed health assistant course during the 2018-2019 academic year, at a medical university in Magway, Myanmar.The condensed health assistant (CHA) course is different from the regular 4-year health assistant course. This 9-month condensed course is intended for the public health staff of the Department of Public Health (DoPH), Ministry of Health and Sports (MoHS), Myanmar. The participants of CHA course started their career as Public Health Supervisor Grade 2 (PHS 2) in DoPH, MoHS. When they have at least 3 years government service, they can sit for the promotion exam for Public Health Supervisor Grade 1 (PHS 1). After 3 years of service as PHS 1, they get another chance to sit for the entrance exam to join the CHA course. Therefore, the age of the students in the CHA course is much older than the students in regular 4-year health assistant course. The participants who attended the lectures at the day of data collection and who gave their written informed consent to participate in the study were recruited. The total number of students attending the 2018 academic year at the university was 802. Among them, 453 students gave their consent to participate in this study. After data cleaning, 41 participants were excluded from the dataset due to incomplete data. Finally, 412 students (277 males and 135 females) were included in the data analysis.” [Methods, Line 149 to 165, Page 6&7]

II. Methods:

Q6: Please provide number of items, scoring method, example questions from each of the three sections.

Authors’ response: Thank you so much for your comment. As suggested, both English and Myanmar version of self-administered questionnaire used in this study was provided to the reviewers in separate files which include number of items, example questions from each of the three sections. No scoring method was used in this study.

III. Results:

Q7: It is suggested to rectify some expressions such as "adverse consequence" to "association", as cross-sectional studies do not provide causal inference to determine reason/consequence. The authors mentioned health and mental health correlates as one of the impacts to test in the Method section, it is suggested to present these variables in Table 3.

Authors’ response: Thank you so much for your comments. As suggested, we revised the “adverse consequence” into “association” throughout the manuscript. [Abstract, Line 32, Page 2], [Introduction, Line 143, Page 6], [Methods: Line 179, Page7; Line 196, Page 8], [Results, Line 245 (Table 3 title), Page 11], and [Discussion: Line 309, Page 16; Line 404, Page 19]

 In this study, to detect the adverse events following cyberbullying victimization, total five questions (inquiring started/increased substance use, difficulties in concentrating and understanding the lectures, suicidal ideation during the last 12 months) were used in the self-administered questionnaire section three, and no scoring method or measurement scale was used (only the answers “Yes, No, Don’t know or No response” were used). And we described this fact as one of the limitations of this cross-sectional study by revising the previous sentence “The extent of the adverse consequences of cyberbullying could not be measured accurately in this study.” into to read as follows; “The extent of the adverse association of cyberbullying could not be measured accurately as no measurement scale or a comparison group was used in this study.” [Discussion, Line 403 to 405, Page 19]

IV. Discussion:

Q8: Some descriptions need citation, eg. paragraph 2 "evidence from previous studies is consistent with this study". Some descriptions need justification, eg. paragraph 2 "periodic screening...and counseling services...are also needed", or move it to the implication part for detailed discussion. 

Authors’ response: Thank you so much for your comments. As suggested, we put citation in the previous paragraph 2 of the discussion section “Cyberbullying has been described as “New bottle but old wine” [11,12]. Many studies have demonstrated that among students, bullying is strongly correlated with poor academic performance or outcomes [24,31-33]. Students who suffered peer bullying received lower grades and, faced academic difficulties and/or worsened academic performance compared with their non-bullied peers [24,25,31-33]. Evidence from previous studies is consistent with this study. Cyberbullying victims were nearly four times more likely to face difficulties in concentrating and understanding lectures compared with non-victims as bullying creates unsafe environment for the students to live in [24]. Cyberbullying is sometimes considered to be more extreme than traditional bullying as it has several unique characteristics that can affect victims anywhere at any time, creating small or large negative impact [13,21,22]. Periodic screening of cyberbullying and counselling services for students who suffered cyberbullying are also needed [13,14,18,28].” into to read as follows; “Cyberbullying has been described as “new bottle but old wine” [15,16]. Many studies have demonstrated that among students, traditional bullying as well as cyberbullying is strongly related to poor academic performance or outcomes [24-26,28,35-37]. Students who suffered peer bullying received lower grades, and faced academic difficulties and/or worsened academic performance compared with their non-bullied peers [24-26,28,29,35-37]. In this context, this study is consistent with the evidence from previous studies [24-26,28,29,35-37]. Cyberbullying victims were nearly four times more likely to face difficulties in concentrating and understanding lectures compared to non-victims, as bullying creates an unsafe environment for the students to live in [28]. Cyberbullying is sometimes considered to be more extreme than traditional bullying, as it has several unique characteristics that can affect victims anywhere, and at any time, creating small or large negative impacts [17,25,26]. Periodic screening of cyberbullying and counselling services for students who suffered cyberbullying are also needed as these interventions are found to be effective to prevent and mitigate adverse outcomes of cyberbullying [17,18,22,32], and Myanmar still lacks such kind of interventions in schools or universities.” [Discussion; Line 322 to 336, Page 16&17]

 According to the suggestion, the justification for the sentence “Periodic screening of cyberbullying and counselling services for students who suffered cyberbullying are also needed [13,14,18,28]” in the previous paragraph 2 of the discussion section was added and changed to read as follows; “Periodic screening of cyberbullying and counselling services for students who suffered cyberbullying are also needed as these interventions are found to be effective to prevent and mitigate adverse outcomes of cyberbullying [17,18,22,32], and Myanmar still lacks such kind of interventions in schools or universities.” [Discussion, Line 333 to 336, Page 16&17]

Q9: The discussion on non-revealing of the cyberbullying experiences is comprehensive but kind of distracted the topic, please amend the logic.

Authors’ response: Thank you so much for your comment. As suggested, the previous sentences regarding non-revealing of the cyberbullying experiences “In this study, only half of the cyberbullying victims told anyone about their experience(s). The low percentage of victims who were willing to tell others about their suffering in this study was consistent with the percentages found in the majority of previous studies [6,11,15-19]. On the contrary, one multi-country study found that more than two-third of the participants shared their experience with others [37]. There might be several important reasons behind less reporting in this study. The main reason may be the victims’ feeling of helplessness to end being attacked or negligence by the adults or authority figures concerning cyberbullying as there is currently no direct law or regulation against cyberbullying in Myanmar [38,39]. However, at the moment, victims can file lawsuits under the Telecommunications Law and the Electronic Transactions Law. Other reasons for less reporting may be the fear of internet access restriction or prohibition by the family or teachers, and facing more cyberbullying attacks if they reported [28]. Therefore, it is important to create and implement awareness raising campaigns and educational programs regarding cyber safety in Myanmar. Legal protection measures against cyberbullying should be formulated and applied in the university setting.” is changed to read as follows; “In this study, only half of the cyberbullying victims told anyone about their experience(s). The low percentage of victims who were willing to tell others about their suffering in this study was consistent with the percentages found in the majority of previous studies [6,15,19-23]. There might be several important reasons behind the less reporting observed in this study. The main reason may be the victims’ feeling of helplessness in the face of being attacked, or negligence by the adults or authority figures concerning cyberbullying, as there is currently no direct law or regulation against cyberbullying in Myanmar [48,49]. Other reasons for less reporting may be the fear of internet access restriction or prohibition by family or teachers, and facing more cyberbullying attacks if they reported [32]. Therefore, it is important to create and implement awareness raising campaigns and educational programs regarding cyber safety in Myanmar. Legal protection measures against cyberbullying should be formulated and applied within the university setting.” [Discussion, Line 337 to 348, Page 17]

Q10: It is suggested to discuss why "no association between amount of time spent and cyberbullying was found in this study".

Authors’ response: Thank you so much for your comment. As suggested, the discussion for ‘no association between amount of time spent and cyberbullying was found in this study’ was added in the previous paragraph 9th of the discussion section “Cyberbullying was found to be associated with the amount of time spent on the internet or social media [6,23,42,43]. The longer the time spent online, the higher the risk of suffering cyberbullying [6,23,42,43]. However, no association between the amount of time spent on social media per day among students and suffering cyberbullying in the past 12 months was found in this study.” that was changed into to read as follows; “Cyberbullying was found to be associated with the amount of time spent on the internet or social media [6,27,54,55]. The longer the time spent online, the higher the risk of suffering cyberbullying victimization [6,27,54,55]. However, no association between the amount of time spent on social media per day among students and suffering cyberbullying victimization in the past 12 months was found in this study. The reason may be due to the difference in the use of common social media among various countries. Majority of participants in this study used Facebook as the most common social media in their daily lives, and cyberbullying attack can occur with one post or comment in Facebook or one message through messenger where the victims do not need to spend much time online.” [Discussion; Line 392 to 400, Page 19]

Newly added references

7. Kowalski RM, Giumetti GW, Schroeder AN, Lattanner MR. Bullying in the digital age: a critical review and meta-analysis of cyberbullying research among youth. Psychol Bull. 2014;140(4):1073-1137. doi:10.1037/a0035618.

8. Siddiqui SA. Cyberbullying and cyber-victimization: from online suicide groups to ‘blue whale’ menace. Indian Pediatr. 2017; 54(12):1056. Available from https://indianpediatrics.net/dec2017/1056.pdf. PubMed PMID: 29317566.

10. Willard NE. Cyberbullying and cyberthreats: responding to the challenge of online social aggression, threats, and distress. Campaign:Research press; 2007. Available from: https://books.google.com.mm/books?hl=en&lr=&id=VyTdG2BTnl4C&oi=fnd&pg=PP7&dq=Cybebullying+and+cyberthreats:+Responding+to+the+challenge+of+online+social+aggression,+threas,+and+distress.+Champaign,+IL:+Research+Press.&ots=u6Ih1Lul6s&sig=1GO4okCOY4BJskbWxyg9ITR8Q&redir_esc=y#v=onepage&q=Cyberbullying%20and%20cyberthreats%3A%20Respondin20to%20the%20challenge%20of%20online%20social%20aggression%2C%20threats%2C%20and20distress.%20Champaign%2C%20IL%3A%20Research%20Press.&f=false

39. Kowalski RM, Limber SE, Agatston PW. Cyberbullying: bullying in the digital age. 2nd ed. Malden, MA: Wiley-Blackwell; 2012. Available from: https://leseprobe.buch.de/images-adb/b0/f1/b0f15512-46cb-4a7d-b8fe-533273e9c564.pdf.

40. Kowalski RM, Giumetti GW, Schroeder AN, Reese HH. Cyber bullying among college students: evidence from multiple domains of college life. Wankel L and Wankel C, editors. Bingley: Emerald Group Publishing Limited; p. 293-321. 

41. Li Q. Cyberbullying in schools: a research of gender differences. Sch Psychol Int. 2006;27(2):157-170. doi:10.1177/0143034306064547. 

42. Ruangnapakul N, Salam YD, Shawkat AR. A Systematic Analysis of Cyber bullying in Southeast Asia Countries. International Journal of Innovative Technology and Exploring Engineering. 2019;8(8S):104-111. ISSN: 2278-3075. Available from: https://www.ijitee.org/wp-content/uploads/papers/v8i8s/H10200688S19.pdf

43. Kwan GC, Skoric MM. Facebook bullying: an extension of battles in school. Computers in human behavior. Comput Human Behav. 2013;29(1):16-25. doi:10.1016/j.chb.2012.07.014.

44. Balakrishnan V. Cyberbullying among young adults in Malaysia: The roles of gender, age and Internet frequency. Comput Human Behav. 2015;46:149-157. doi:10.1016/j.chb.2015.01.021.

45. Musikaphan W. A study of cyber-bullying in the context of Thailand and Japan. Nakhon Pathom: National Institute for Child and Family Development, Mahidol University, Thailand. 2009.

46. Safaria T. Prevalence and impact of cyberbullying in a sample of Indonesian junior high school students. Turkish Online Journal of Educational Technology. 2016;15(1):82-91. ISSN: EISSN-1303-6521. Available from: https://files.eric.ed.gov/fulltext/EJ1086191.pdf.

52. Cénat JM, Smith K, Hébert M, Derivois D. Cybervictimization and suicidality among French undergraduate students: a mediation model. J Affect Disord. 2019;15(249):90-95. doi:10.1016/j.jad.2019.02.026.

53. Chang Q, Xing J, Ho RT, Yip PS. Cyberbullying and suicide ideation among Hong Kong adolescents: the mitigating effects of life satisfaction with family, classmates and academic results. Psychiatry Res. 2019;1(274):269-273. doi:10.1016/j.psychres.2019.02.054.

Additional revision

(1) We change the previous reference numbers 7 to 34 into 11 to 38, the previous reference number 35 to 47, the previous reference number 36 to 9, the previous reference numbers 38 to 41 into 48 to 51, and the previous reference number 42 and 43 into 54 and 55 to cover all three reviewers’ comments.

(2) The previous reference number 37 is removed to cover Reviewer 3’s comment.

---

## [Decision Letter · Decision Letter 1]

12 Dec 2019

Assessing risk factors and impact of cyberbullying victimization among university students in Myanmar: A cross-sectional study

PONE-D-19-16609R1

Dear Dr. Saw,

We are pleased to inform you that your manuscript has been judged scientifically suitable for publication and will be formally accepted for publication once it complies with all outstanding technical requirements.

With kind regards,

Siyan Yi, MD, MHSc, PhD

Academic Editor

PLOS ONE

Additional Editor Comments (optional):

Reviewers' comments:

Reviewer's Responses to Questions

**Comments to the Author**

1. If the authors have adequately addressed your comments raised in a previous round of review and you feel that this manuscript is now acceptable for publication, you may indicate that here to bypass the “Comments to the Author” section, enter your conflict of interest statement in the “Confidential to Editor” section, and submit your "Accept" recommendation.

Reviewer #1: All comments have been addressed

Reviewer #2: All comments have been addressed

2. Is the manuscript technically sound, and do the data support the conclusions?

Reviewer #1: Yes

Reviewer #2: Yes

3. Has the statistical analysis been performed appropriately and rigorously? 

Reviewer #1: Yes

Reviewer #2: Yes

4. Have the authors made all data underlying the findings in their manuscript fully available?

Reviewer #1: Yes

Reviewer #2: Yes

5. Is the manuscript presented in an intelligible fashion and written in standard English?

Reviewer #1: Yes

Reviewer #2: Yes

6. Review Comments to the Author

Reviewer #1: The authors have adequately addressed most of the reviewer recommendations. There was a great effort on behalf of the author(s) in improving the paper.

Reviewer #2: (No Response)

7. PLOS authors have the option to publish the peer review history of their article (what does this mean?). If published, this will include your full peer review and any attached files.

Reviewer #1: Yes: Dr. Goran Livazović, Associate Professor, Department for Pedagogy, Faculty of Humanities and Social Sciences in Osijek, Republic of Croatia

Reviewer #2: Yes: Shahid Akhtar Siddiqui

Assistant Professor

Department of Pediatrics

MLN Medical College, Allahabad. U.P. 211002

sha.akht@gmail.com

---

## [Editor Report · Acceptance letter]

31 Dec 2019

PONE-D-19-16609R1 

Assessing risk factors and impact of cyberbullying victimization among university students in Myanmar: A cross-sectional study 

Dear Dr. Saw:

I am pleased to inform you that your manuscript has been deemed suitable for publication in PLOS ONE. Congratulations! Your manuscript is now with our production department. 

With kind regards,

on behalf of

Dr. Siyan Yi 

Academic Editor

PLOS ONE